# Interpretable Table Question Answering via Plans of Atomic Table Transformations

## Abstract

Interpretability for Table Question Answering (Table QA) is critical, particularly in high-stakes domains like finance or healthcare. While recent Large Language Models (LLMs) have improved the accuracy of Table QA models, their explanations for how answers are derived may not be transparent, hindering user ability to trust, explain, and debug predicted answers, especially on complex queries. We introduce Plan-of-SQLs (**POS**), a novel method specifically crafted to enhance interpretability by decomposing a query into simpler sub-queries that are sequentially translated into SQL commands to generate the final answer. Unlike existing approaches, **POS** offers full transparency in Table QA by ensuring that every transformation of the table is traceable, allowing users to follow the reasoning process step-by-step. Via subjective and objective evaluations, we show that **POS** explanations significantly improve interpretability, enabling both human and LLM judges to predict model responses with 93.00% and 85.25% accuracy, respectively. **POS** explanations also consistently rank highest in clarity, coherence, and helpfulness compared to state-of-the-art Table QA methods such as Chain-of-Table (Wang et al., 2023) and DATER (Ye et al., 2023). Furthermore, **POS** demonstrates high accuracy on Table QA benchmarks (78.31% on TabFact and 54.80% on WikiTQ with GPT3.5), outperforming methods that rely solely on LLMs or programs for table transformations, while remaining competitive with hybrid approaches that often trade off interpretability for accuracy.

## 1 Introduction

Table QA models enable users to quickly retrieve the desired information from large and complex tables. Recently, LLMs have revolutionized the landscape of Table QA literature with state-of-the-art performance on a wide range of Table QA benchmarks. These models are highly accurate and sometimes are deemed interpretable (Ye et al., 2023; Cheng et al., 2023; Wang et al., 2023; Nahid & Rafiei, 2024) but the researchers provide no data to support the claims of interpretability.

Answers from Table QA models can be explained via intermediate tables and operations, offering a step-by-step walk-through of the reasoning process. For example, Chain-of-Table (CoTable) (Wang et al., 2023) progressively transforms the input table, through several function calls (predicted by LLMs), into a simplified table to be presented to the LLM to ask for the final answer—Fig. 1(**c**).

However, there are two main challenges with the interpretability of current LLM-based Table QA models. First, the reasoning becomes increasingly uninterpretable as query complexity grows—e.g., when a table contain numerous rows and columns or when the question involves multiple conditions. In Fig. 1(**c**), CoTable decides to select five rows using a function call of **f_select_row(**2, 3, 4, 5, 9**)**. Yet, there is no explanation for why these rows are chosen. Second, the final step, where the answer is generated, still relies on the black-box reasoning of a model—leaving users uninformed as to why the final answer was arrived at—an issue commonly observed in many works (Jiang et al., 2023; Wang et al., 2023; Ye et al., 2023; Nahid & Rafiei, 2024; Abhyankar et al., 2024; Wu & Feng, 2024). This introduces another layer of opacity in the reasoning process of Table QA models.

To address these interpretability challenges, we propose Plan-of-SQLs (**POS**)—a novel method that breaks down the original query into simple natural language sub-queries, which are easily converted into SQL commands and understandable by humans. For example, steps like **Select rows where** or **Select column** are translated into SQL commands that are executed *sequentially*. This approach ensures that each transformation is explicitly rational, thus preventing the model from arbitrarily

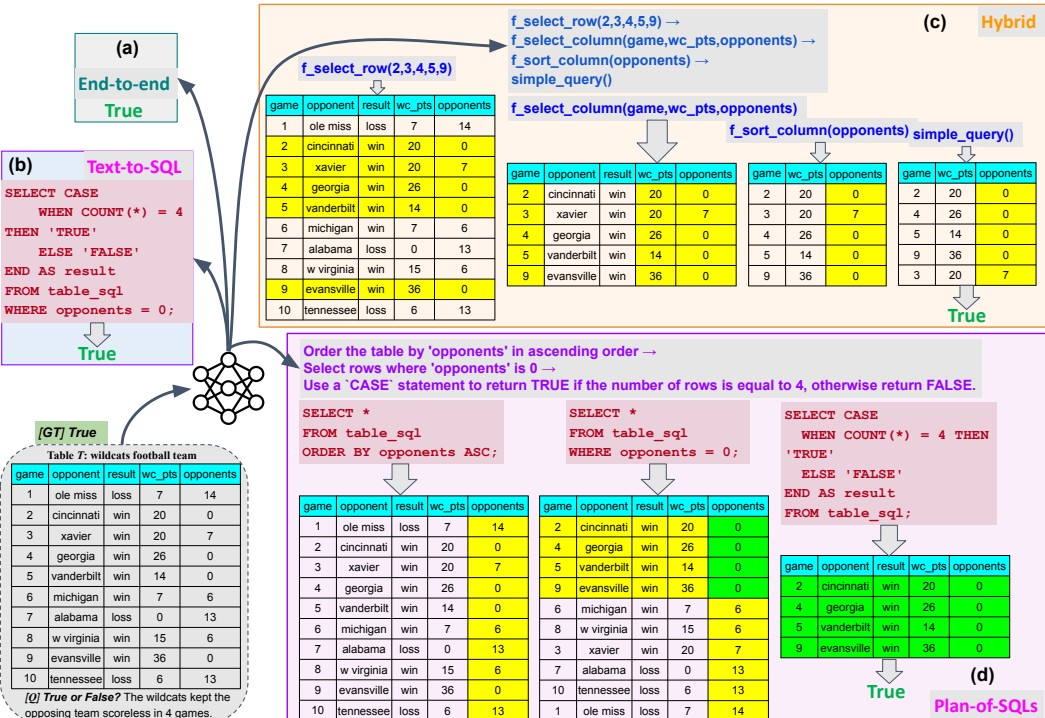

Figure 1: Different approaches to Table QA correctly answering the question as **True**. **(a)** End-to-End: Relies entirely on an LLM to answer the question directly, leaving no room for users to understand the prediction. **(b)** Text-to-SQL: Generates a SQL command to solve the query, requiring domain expertise to comprehend and becoming unintelligible when the query becomes complex. **(c)** CoTable: Performs planning with atomic functions and executes sequentially to arrive at the final answer. However, function arguments are not justified, and the final answer again depends on the LLM's opaque reasoning. In contrast, **(d)** Plan-of-SQLs or **POS** (our proposal): Plans in natural language, making each step simple and understandable. Each step is then converted into a SQL command, sequentially transforming the input table end-to-end to produce the final answer.

selecting irrelevant data as seen in Fig. 1**(c)**. Furthermore, our method directly addresses the opacity in the final step of existing Table QA models as shown in Fig. 1**(a)** & **(c)** by generating the final answer through the same transparent, SQL-based process like Text-to-SQL (Rajkumar et al., 2022) in Fig. 1**(b)**. Yet, our method improves on Text-to-SQL that requires domain expertise to comprehend SQLs and often produces unexecutable programs for complex queries (Shi et al., 2020).

To investigate the interpretability of **POS**, we conduct both subjective and objective experiments. We present the judges (humans and LLMs) with explanations from both our method and those generated by state-of-the-art QA models, such as Text-to-SQL (Rajkumar et al., 2022), DATER (Ye et al., 2023), and CoTable (Wang et al., 2023)—studying which explanations are preferred or effective for the judges in predicting model responses. We also evaluate **POS**'s Table QA accuracy on the two standard benchmarks: TabFact (Chen et al., 2020) and WikiTQ (Pasupat & Liang, 2015). Our main contributions are summarized as follows:

- We introduce **POS** (Fig. 2)—a Table QA method specifically designed for interpretability, decomposing complex queries into atomic natural-language sub-queries, which are then translated into SQL commands to sequentially transform input table into final answer.

- Via subjective and objective evaluations using humans and LLM judges (Tab. 1 & 2), we show that **POS** explanations (i) significantly improve judges' accuracy in predicting model behavior up to 93.00% and and (ii) receive the best rankings in terms of clarity, coherence, and helpfulness in understanding model's reasoning process. Ours is also the first work showing strong correlations between human and LLM judges in evaluating Explainable AI (XAI) methods.

- In Tab. 3, our **POS** achieves 78.31% accuracy on TabFact and 54.8% on WikiTQ with GPT3.5, outperforming other program-based approaches and those using LLM only. Furthermore, our method achieves competitive performance when compared to state-of-the-art Table QA models that employ hybrid approaches—combinations of program-based and LLM reasoning. These hybrid models, while powerful, often sacrifice on interpretability, as they still rely on the black-box reasoning of LLMs for table transformations.

## 2 RELATED WORK

### 2.1 DECOMPOSING COMPLEX INPUT QUERIES IN TABLE QA

LLM-based Table QA models have improved performance by decomposing complex input queries into sub-problems (Ye et al., 2023; Nahid & Rafiei, 2024) or step-by-step reasoning (Wang et al., 2023; Wu & Feng, 2024; Abhyankar et al., 2024). This breakdown effectively addresses the compositionality gap, a challenge where models can solve all sub-problems but struggle to combine them into a coherent solution (Press et al., 2023). However, these methods often rely on *complex table transformations*—i.e., selecting a sub-table from the input table based on complex reasoning steps (Ye et al., 2023; Nahid & Rafiei, 2024; Wu & Feng, 2024; Abhyankar et al., 2024), which are prone to errors regarding which table entries to select (refer to Appendix F for examples of hallucination in sub-table selections). For example, DATER (Ye et al., 2023) in Fig. 12 selects a sub-table from the original table based on the statement. However, the inclusion of row 3 is illogical and does not contribute to a valid answer. Our approach mitigates this issue by leveraging *a sequence of simple program-based table transformations*. Each transformation is constrained to be easily executable and atomic, such as a simple **Select rows where opponents is 0** in Fig. 1(**d**)—clause with only one condition and one variable, ensuring clarity and mitigating the hallucination problem.

Closest to our work is CoTable (Wang et al., 2023), which uses predefined **atomic functions**, like (Chen et al., 2020; Nan et al., 2022; Mouravieff et al., 2024), to transform intermediate tables. However, it still relies on the black-box reasoning of the model, particularly when adding new columns, generating function arguments, or generating the final answer— Fig. 1(**c**). Meanwhile, our method leverages *atomic **natural-language** steps* that are both human-comprehensible and easily convertible into SQLs. The SQL commands are then sequentially applied to the tables, ensuring transparency throughout table transformations and answer generation.

### 2.2 PROGRAM-AIDED TABLE TRANSFORMATIONS

Program-aided table transformations play a crucial role in processing tabular queries. Using languages like SQL (Nahid & Rafiei, 2024; Ye et al., 2023; Cheng et al., 2023) or Python (Cheng et al., 2023; Chen et al., 2020) offers two main advantages over LLMs. First, they enable transparent, rule-based transformations, offering greater traceability and interpretability compared to the opaque generation of LLMs. Second, they are designed for efficient handling of large-scale, complex data operations, making them more reliable and cost-effective than LLM-based methods (i.e., inputting the whole large tables into LLMs is inefficient and erroneous (Chen, 2023; Wang et al., 2023)).

Our work joins a growing body of literature that harnesses program-aid table transformations for Table QA but using SQL commands exclusively, much like Text-to-SQL (Rajkumar et al., 2022). To our knowledge, only two methods in Table QA literature—LPA (Chen et al., 2020) using Python-Pandas and Text-to-SQL—address Table QA queries using program-based operations end-to-end. Yet, since Text-to-SQL generates a single SQL command for the entire task, it requires a highly powerful Text-to-SQL converter and is prone to hallucinations (Shi et al., 2020). Meanwhile, LPA constructs a single Python-Pandas program to represent the entire query, which can lead to complexity and potential errors due to the challenges of synthesizing accurate programs in one step. By breaking down queries into multiple SQL commands, we eliminate the need for powerful Text-to-SQL models while also achieving superior performance (see Tab. 3—decomposing queries into simple SQL operations significantly improves accuracy over Text-to-SQL and LPA on TabFact).

### 2.3 INTERPRETABILITY FOR TABLE QA

Interpretability is a critical aspect of Table QA models, especially when they are applied in high-stakes applications. However, existing Table QA models often only provide limited explanations, typically confined to indicating row indices or column names involved in the reasoning process— Fig. 1(**c**), offering surface-level reasoning without deeper context (Wang et al., 2023; Ye et al., 2023). This leaves users with high-level overviews rather than detailed insights into how specific data points contribute to the final answer.

In contrast, our method (**POS**) advances interpretability in Table QA by providing natural-language, step-by-step explanations that are directly tied to programmatic operations, as shown in Fig. 1(**d**). Each step corresponds to a simple SQL command that is both human-understandable and machine-executable. **POS** also presents attribution maps over intermediate tables, indicating exactly which

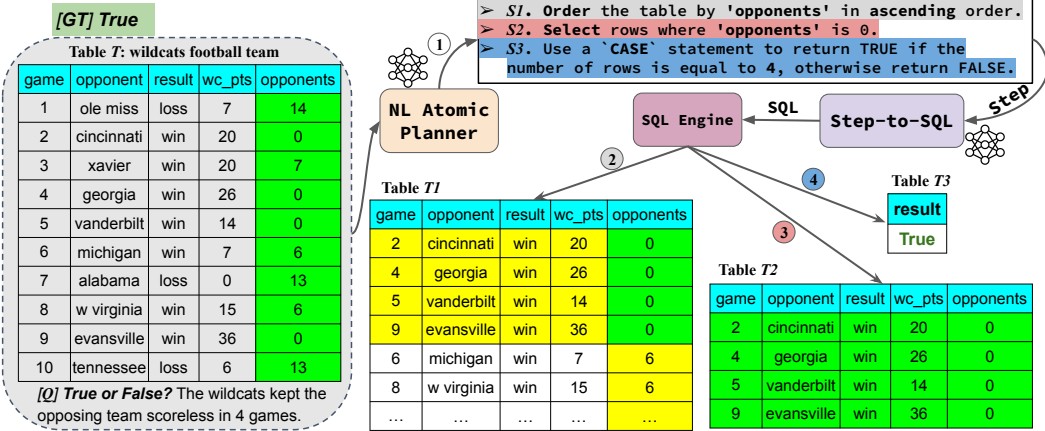

Figure 2: Illustration of Plan-of-SQLs (**POS**). ①The NL Atomic Planner takes $(T, Q)$ as input and generates a step-by-step plan in natural language to answer $Q$. ②**Step-to-SQL** takes $(T, S1)$ as input and converts $S1$ to SQL to sort the table (transform $T \rightarrow T1$). ③**Step-to-SQL** takes $(T1, S2)$ as input and converts $S2$ to SQL to select relevant rows (transform $T1 \rightarrow T2$). ④**Step-to-SQL** takes $(T2, S3)$ as input and converts $S3$ to return the final answer (transform $T2 \rightarrow T3$).

cells are used for the final prediction. This design allows users to follow the reasoning process at a deeper level. Finally, unlike prior work that relies on the black-box reasoning of LLMs to generate the final answer (Ye et al., 2023; Wang et al., 2023; Nahid & Rafiei, 2024), our approach generates the answer through a simple, transparent SQL command.

In evaluating interpretability, we join a long line of works testing machine explanations on human users (Adebayo et al., 2020; Nguyen et al., 2021; Kim et al., 2022; Taesiri et al., 2022; Colin et al., 2022; Chen et al., 2023; Nguyen et al., 2024a;b). Yet, to the best of our knowledge, this is the first work to test explanations on human and LLM judges in the context of LLM-based Table QA.

# 3 **POS**: INTERPRETABLE TABLE QA

**Problem Formulation.** In Table QA, each sample can be represented as a triplet $(T, Q, A)$, where $T$ is a table, $Q$ is a natural language question or statement about the table, and $A$ is the answer. The goal of Table QA is to predict the answer $A$ given the query $Q$ and the table $T$. To achieve this, our method decomposes $Q$ into smaller sub-queries (atomic steps), followed by converting them into SQL commands, and applying these commands sequentially to the table to arrive at the answer $A$.

**Grounded Table QA.** Our method processes Table QA queries entirely through SQL commands, with table transformations executed offline and no access to external knowledge bases[1]. Thus, we assume that all information necessary to answer the query is contained within the original table, following the definition of grounded QA as termed in the literature (Lei et al., 2018).

However, we observe that grounded QA is not established in popular Table QA benchmarks—i.e., the ground-truth is not always present in the input table. For instance, 18.9% of queries in WikiTQ are insufficiently expressive using SQL to provide the correct answer and require external knowledge (Shi et al., 2020). This problem contributes to the performance gap between our SQL-based method vs. hybrid approaches on WikiTQ (see Tab. 3). We wish to clarify that our method is designed with a focus on interpretability rather than accuracy alone.

**Atomicity.** In **POS**, we define an atomic step as a simple, minimal operation that can be directly translated into a single SQL command. Specifically, an atomic SQL command (i) contains at most one condition in the WHERE clause; (ii) uses at most one variable or column in that condition. By restricting each step to be atomic in this way, we ensure that the SQL commands are straightforward and less prone to errors during Step-to-SQL translation and execution. This also allows us to better explain the reasoning process of a model to users. Examples of atomic steps in Appendix I.1.

---

[1]For unexecutable samples, we fallback to end-to-end QA, similar to (Cheng et al., 2023; Kong et al., 2024).

### 3.1 Generating Natural-Language Atomic Plans

We perform planning in natural language, which aligns closely with LLM capabilities (Huang et al., 2022). This also makes the planning process more interpretable to humans, as each step is expressed in clear, understandable terms rather than function calls, whose motivations and arguments are *not* explained to users. We study the importance of natural-language planning in Appendix C.

In Fig. 2–①, the **Natural-Language (NL) Atomic Planner** takes $(T, Q)$ as input and converts $Q$ into a plan of sub-queries, referred to as **atomic steps**. The generated plan outlines the sequence of operations needed to arrive at the answer $A$. Below is the prompt we use for planning on TabFact:

---

**Prompt to Generate Natural-Language Atomic Plans**

[**In-context Planning Examples**]

[**Input Table**]

[**Statement**]

Let's develop a step-by-step plan to verify if the given Statement is TRUE or FALSE on the given Table! You MUST carefully analyze the Statement and comprehend it before writing the plan!

**Plan Steps:** Each step in your plan should be very atomic and straightforward, ensuring they can be easily executed or converted into SQL. You MUST make sure all conditions are checked properly in the steps.

**Step order:** The order of steps is crucial! You must ensure the orders support the correct information retrieval and verification! The next step will be executed on the output table of the previous step.

For comparative or superlative Statements, you should order the table accordingly before selecting rows. This ensures that the desired comparative or superlative data is correctly retrieved.

**Plan:**

---

### 3.2 Executing Atomic Plans with SQL Commands

After generating a natural-language plan in Sec. 3.1, the next step is to operationalize it by converting each atomic step into an executable SQL. This translation is fundamental to **POS**, bridging the gap between high-level natural language reasoning and reliable, transparent table transformations.

#### 3.2.1 Step-to-SQL: Converting Atomic Steps to SQL Commands

Leveraging the versatile capabilities of LLMs as Text-to-SQL converters (Hong et al., 2024), we translate each step into its corresponding SQL query. By ensuring that each step is atomic and straightforward (Sec. 3.1), we mitigate the need for complex Text-to-SQL translations. The conversion process involves crafting a prompt for the LLM that includes the current state of the table, the specific natural-language step to be performed, and any constraints to guide the LLM in generating executable SQL commands. We prompt **Step-to-SQL** to convert a NL step into SQL as follows:

---

**Prompt to Convert Natural-Language Steps to SQLs**

[**In-context Step-to-SQL Examples**]

[**Input Table**]

Write a SQL command that: [**natural_language_step**]
Constraints for your SQL:

1. If using SELECT COUNT(*), SUM, MAX, AVG, you MUST use AS to name the new column. If adding new columns, they should be different than existing columns.
2. Your SQL command MUST be compatible and executable by Python sqlite3 and Pandas.

---

#### 3.2.2 Sequential Execution of SQL Commands

Once each step is translated into SQL, we execute it using a lightweight SQL engine called `sqlite3` (Muddana & Vinayakam, 2024). The execution proceeds sequentially: the output of one SQL command becomes the input for the next, effectively chaining together the transformations specified by the generated plan (Fig. 2–②–③–④). In contrast to end-to-end and hybrid QA methods that rely on the black-box reasoning of LLMs for final-answer generation— Fig. 1**(a)** & **(c)**, **POS** maintains transparency throughout by SQLs only. We provide all details of prompt engineering for **POS** in Appendix H.

Figure 3: Generating attributions maps for **POS**. Column idx is added to track row attribution.

## 3.3 GENERATING EXPLANATIONS FOR LLM-BASED TABLE QA MODELS

Despite the importance of interpretability, there has been a noticeable gap in the visualization of explanations for LLM-based Table QA. Previous works have largely focused on improving accuracy and efficiency without providing insights into the reasoning processes (Wu & Feng, 2024; Kong et al., 2024). Motivated by that, we propose a pipeline to generate explanations for Table QA models by leveraging the intermediate information given during the execution of operations. Our approach highlights relevant parts of the intermediate tables to create *attribution maps* (Montavon et al., 2018), which illustrate how different input features contribute to the prediction.

### 3.3.1 ATTRIBUTION MAPS

During the execution of each SQL command, we perform the following steps:

- **Adding the tracking index column:** Before executing an SQL, we add a tracking index column to the current table. This column contains the original row indices from the initial table– Fig. 3(**a**).
- **Executing the SQL command:** An SQL command is executed on the table with the tracking index column, producing a modified table– Fig. 3(**b**).
- **Identifying selected rows:** After execution, we use the tracking index column to identify which rows have been selected or filtered by the SQL command– Fig. 3(**c**).
- **Identifying selected columns:** We parse the SQL command to extract the columns involved in the operation– Fig. 3(**d**) (more details in Appendix E).
- **Visualizing an attribution map:** The previous steps allow us to generate an attribution map for the initial table– Fig. 3(**e**). The index column is also drop here.

Since both rows and columns can be attributed within an operation, **POS** offers a distinct advantage over previous works (Ye et al., 2023; Wang et al., 2023)—accurately attributing responsible cells for each transformation. For example, when a SQL command includes a condition that requires a cell to match a specific value or range (e.g., WHERE opponents = 0), we can determine which cells in the opponents column satisfy this condition and are thus responsible for the answer– Fig. 3(**e**).

### 3.3.2 CHAIN-OF-HIGHLIGHTED-TABLE EXPLANATIONS

At each stage of the table transformation process, we visualize attribution maps over the intermediate tables, emphasizing the data selected or filtered in the current operation. Rows and columns containing relevant data for the operation are yellow-highlighted, while cells matching the specific condition at this step are green-highlighted. Using the information obtained from the plan execution and attribution maps, we combine the three components: (1) intermediate tables; (2) attribution maps; and (3) step description ; to make an explanation shown to users. We present the explanation in a *chain of highlighted intermediate tables* for Table QA models, helping users visually follow the sequence of transformations and understand how each step contributes to the final answer. Please refer to Appendix A for explanations from Text-to-SQL, DATER, CoTable, and **POS**.

## 4 EXPERIMENTS

We use two widely-used Table QA benchmarks: TabFact (Chen et al., 2020) and WikiTQ (Pasupat & Liang, 2015) in our experiments.

**TabFact** is a fact verification dataset where each statement on a table should be labeled with either TRUE or FALSE (see Fig. 1). We use the cleaned TabFact dataset provided by Wang et al. (2023) and evaluate model binary classification accuracy on the test-small set of 2,024 samples.

**WikiTQ** involves complex question answering, where the task is to answer a question written by human annotators by retrieving or inferring information from an input table. We use the dataset and evaluation scripts provided by Ye et al. (2023) and evaluate model denotation accuracy (whether the predicted answer is equal to the ground-truth answer) on the standard test set of 4,344 samples.

### 4.1 EVALUATING EXPLANATION METHODS FOR TABLE QA

We follow the two evaluation settings proposed by Doshi-Velez & Kim (2017), assessing XAI methods both subjectively and objectively, using human users and LLMs (we refer to both as XAI judges).

**Subjective evaluation:** XAI judges are presented with explanations generated only from samples where all methods either produce correct or incorrect results then asked to rank them based on perceived quality; a.k.a. **Preference** task (Ramaswamy et al., 2023; Yang et al., 2024). By using relative rankings, we directly compare the methods in terms of clarity, coherence, and helpfulness in understanding the model's reasoning (see the task setup in Appendix G).

**Objective evaluation:** XAI judges are provided with an explanation and an input then tasked with accurately predicting the model's output, regardless of the ground-truth—the **Forward Simulation** task (Doshi-Velez & Kim, 2017; Hase & Bansal, 2020; Chen et al., 2022) (see Appendix G for LLM-as-a-Judge setup and Appendix K for human study setup).

**Baselines** We select Text-to-SQL (Rajkumar et al., 2022), DATER (Ye et al., 2023),and CoTable (Wang et al., 2023) as baseline XAI methods w.r.t their state-of-the-art performance in Table QA, interpretability, and reproducibility. We present baseline details in Appendix A.

**Visualizations** For our experiments, we use the TabFact (Chen et al., 2020) dataset, running each method across the entire test set of 2,024 samples with `gpt-3.5-turbo-16k-0613`. We visualize the explanations for the executable samples by utilizing the intermediate information from each method. In total, we generate 1,340 visualizations for Text-to-SQL, 2,024 for DATER, 2,024 for CoTable, and 1,952 for **POS**.

#### 4.1.1 EVALUATING EXPLANATIONS WITH HUMAN USERS

**Motivation** Human judges are the gold standard for assessing explanations, as they are the ones ultimately interacting with AI models via explanation interfaces (Doshi-Velez & Kim, 2017). We aim to study how explanations help humans in predicting model behaviors in Forward Simulation.

**Human Judges** We recruit 32 volunteers for our study on forward simulation, all of whom are computer science undergraduate, master's, or Ph.D. students. In each session, a user can choose one of the four explanation methods and is asked to complete 10 samples. Each user can complete as many sessions as they wish. In total, we gather 800 responses, with each method receiving around 200 responses. Please refer to Appendix K for the complete flow of our human study interface.

#### 4.1.2 EVALUATING EXPLANATIONS WITH LLM-AS-A-JUDGE

**Motivation** The use of LLMs trained on human-alignment data (Ouyang et al., 2022) as judges has been garnering attention due to their strong correlation with human judgments. In two-alternative forced choice (2AFC) tasks, models like GPT-4 achieve strong agreement with humans (Dubois et al., 2024; Zheng et al., 2023). Further, LLM-powered judges like G-Eval (Liu et al., 2023) also demonstrate strong correlations with humans in natural-language generation evaluation metrics. This evidence makes LLMs promising, scalable judges for evaluating explanation quality, particularly in tasks like Table QA, where the information is still text-based but structured. In this work, we leverage LLM judges for both Preference and Forward Simulation.

Table 1: Forward Simulation accuracy (%) of LLM and human judges for XAI methods.

| XAI method | Text-to-SQL | DATER | CoTable | **POS** (Ours) |
|---|---|---|---|---|
| GPT-4o-mini | 65.67 | 73.57 | 76.53 | **81.61** |
| GPT-4o | 73.73 | 78.21 | 79.55 | **85.25** |
| GPT-4 | 75.15 | 80.04 | 79.99 | **84.89** |
| Human | 83.68 | 86.50 | 84.29 | **93.00** |

**LLM Judges**   Motivated by previous works showing the effectiveness of OpenAI's GPT models as reliable judges (Zheng et al., 2023; Liu et al., 2023; Dubois et al., 2024), we utilize 3 OpenAI's models: `gpt-4-turbo-2024-04-09`, `gpt-4o`, and `gpt-4o-mini` to judge Table QA explanations. Please refer to Appendix G for detailed setup of LLM-as-a-Judge experiments.

### 4.1.3 EXPERIMENTAL RESULTS

**POS explanations are most effective in forward simulation**   Tab. 1 shows that **POS** explanations significantly boost both human and LLM judges' accuracy in predicting the model's output. Specifically, human judges achieve 93.00% with **POS** explanations, outperforming other methods such as DATER (86.50%) and CoTable (84.29%). Similarly, across all LLM judges, **POS** consistently yields the highest accuracy, with improvements ranging from 5% − 6% over the next best XAI method. This consistent superiority suggests that **POS** explanations provide more informative insights into the model's reasoning process, thereby facilitating better effectiveness.

**Text-to-SQL explanations are least effective in forward simulation**   Text-to-SQL consistently results in the lowest accuracy among both human and LLM judges. Human judges achieve an accuracy of 83.68% with Text-to-SQL explanations, which is notably lower than their performance with **POS** (93.00%). Similarly, LLM judges show the poorest performance with Text-to-SQL, with accuracy ranging from 65.67% to 75.15%. This suggests that while Text-to-SQL provides a precise logical sequence in the form of SQL queries, its technical nature and requirement for expertise make it less effective for the task.

**Human judges outperform LLMs in forward simulation**   Tab. 1 also reveals that human judges surpass LLMs in accurately predicting the model's outputs across all explanations. For instance, with **POS** explanations, humans achieve an accuracy of 93.00%, while the highest accuracy among LLM judges is 85.25% from `gpt-4o`. This performance gap indicates that humans possess a superior ability to interpret explanations and contextual nuances that LLMs might overlook.

**POS explanations are considered best-quality by LLM judges**   In Tab. 2, **POS** explanations consistently receive the best rankings across all LLM judges. Specifically, our proposed explanations achieve average rankings of 1.55, 1.01, and 1.33 from `GPT-4o-mini`, `GPT-4o`, and `GPT-4` respectively, substantially outperforming CoTable, DATER, and Text-to-SQL. This shows that **POS** explanations are perceived by the judges as providing the best clarity, coherence, and helpfulness in understanding the model's reasoning process (see qualitative definitions in Appendix G).

**Preference rankings strongly correlate with Forward Simulation accuracy**   Using Tab. 1 & Tab. 2, we perform a correlation analysis between Preference rankings vs. Forward Simulation accuracy to investigate if better qualitative assessments correlate with improved quantitative measures. Surprisingly, we find a strong positive correlations across explanation methods.

Since lower rankings in Preference indicate better explanations, we invert the rankings for correlation analysis to align higher accuracy with better preference. For `GPT-4o-mini`, we observe a Pearson correlation coefficient of $r=0.9685$ (p-value: 0.0315), indicating a significant positive relationship. Similar positive correlations are found for `GPT-4o` ($r=0.9333$, p-value: 0.0667) and `GPT-4` ($r=0.8047$, p-value: 0.1953). The overall correlation coefficient across all models is

Table 2: Relative rankings for XAI methods given by LLM XAI judges. Lower values indicate better rankings (1 = best, 4 = worst). For fair comparisons, we perform Preference Ranking on $n$=707 samples where all four methods are executable and either all generate correct answers or all generate incorrect answers.

| XAI method | Text-to-SQL | DATER | CoTable | **POS** (Ours) |
|---|---|---|---|---|
| GPT-4o-mini | 3.95 | 2.75 | 1.75 | **1.55** |
| GPT-4o | 3.60 | 3.35 | 2.04 | **1.01** |
| GPT-4 | 3.33 | 3.36 | 1.98 | **1.33** |

$r$=0.7865 (p-value: 0.0024), confirming a robust positive correlation. These findings suggest that the perceived quality of explanations—as measured by preference rankings—is predictive of their effectiveness in helping judges accurately predict the model's outputs.

Table 3: Accuracy (%) for TabFact and WikiTQ using GPT3.5 (`gpt-3.5-turbo-16k-0613`). "Breakdown" indicates whether queries are decomposed into sub-problems (Fig. 2–①). "Transformed by" refers to whether intermediate tables are transformed by an LLM or a program (Fig. 2–②–③). "Answered by" specifies whether the final answer is generated by an LLM or a program(Fig. 2–④). LLM-only approaches provide the final answer without table transformations.

| Method | Accuracy (%) | | Breakdown | Tables transformed by | Final answer by |
| --- | --- | --- | --- | --- | --- |
| | TabFact | WikiTQ | | | |
| End-to-End QA | 70.45 | 51.84 | ✗ | - | LLM |
| Few-Shot QA | 71.54 | 52.56 | ✗ | - | LLM |
| Chain-of-Thought (Wei et al., 2022) | 65.37 | 53.48 | ✗ | - | LLM |
| Binder (Cheng et al., 2023) | 79.17 | 56.74 | ✓ | LLM + Program | Program |
| Dater (Ye et al., 2023) | 78.01 | 52.81 | ✓ | Program | LLM |
| CoTable (Wang et al., 2023) | **80.20** | 59.90 | ✓ | Program | LLM |
| TableSQLify (Nahid & Rafiei, 2024) | 79.50 | **64.70** | ✗ | Program | LLM |
| Text-to-SQL (Rajkumar et al., 2022) | 64.71 | 52.90 | ✗ | Program | Program |
| LPA (Chen et al., 2020) | 68.90 | - | ✓ | Program | Program |
| **POS** (Ours) | 78.31 | 54.80 | ✓ | Program | Program |

## 4.2 EVALUATING TABLE QA PERFORMANCE

**Baselines**   We compare **POS** with several baseline methods, categorizing them into three groups based on **how table transformation and answer generation are performed**: LLM-only, program-only, and hybrid approaches. We present details for baselines in Appendix B. Unless otherwise noted, the LLM used in our experiments is `gpt-3.5-turbo-16k-0613`, with a `temp` value set to 0 and `top-p` value of 1 for sampling.

**Results**   **POS** achieves 78.31% accuracy on TabFact and 54.8% on WikiTQ, outperforming LLM-only methods, such as End-to-End QA, Few-Shot QA, and Chain-of-Thought. In addition, **POS** demonstrates significant improvements over program-only methods. For instance, on TabFact, our method improves accuracy by +13.6 pts over Text-to-SQL and +9.41 pts over LPA.

**POS** performs competitively to hybrid approaches on TabFact. However, on WikiTQ, our performance still lags behind the state-of-the-art. This is primarily because our method processes Table QA queries entirely through SQL commands, with table transformations executed offline and no access to external knowledge bases (see Sec. 3). We present an ablation study for **POS** on three key components: Atomicity, Natural-Language Planning, and Step-to-SQL Conversion in Appendix C.

## 5 CONCLUSION AND DISCUSSION

**Limitations**   First, in Tab. 3, we observe that a subset of samples cannot be processed end-to-end with our method (9.8% for TabFact and 27.8% for WikiTQ using `gpt-3.5-turbo-16k-0613`). In such cases, we fallback to an end-to-end question-answering approach, directly querying LLMs for the final answer, similar to (Cheng et al., 2023; Kong et al., 2024). Second, **POS** relies on exact matches between the query and the input table. Although we have incorporated soft-matching techniques using SQL's `LIKE` function, certain cases—such as a query with "thomas børn" and a table entry with "thomas born"—still result in failure to identify the relevant information (see Fig. 18).

**Discussion**   We introduce Plan-of-SQLs (**POS**), a novel approach specifically designed to improve the interpretability of Table QA models. Our findings highlight two key advantages of **POS**. First, it addresses a common limitation in current Table QA literature—the lack of transparency—by making every transformation step understandable. Second, **POS** improves human ability to predict model behaviors, as shown by the high accuracy in Forward Simulation, which suggests that explanations provided by **POS** are not just intuitive but also actionable. Notably, we observe that most of the **POS**'s errors are due to the poor planning capabilities of LLMs, rather than issues with Step-to-SQL translation (see an error analysis in Appendix J). We expect that as LLMs continue improving in planning, **POS** will become more accurate in QA while retaining its current level of interpretability.

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

APPENDIX

## A  BASELINE XAI METHODS FOR TABLE QA

In this section, we present visual explanations for Table QA models, which help bridge the gap between model behavior and human understanding. Each visualization provides insights into how the model interprets the input table, highlighting the key information used in its reasoning process.

We showcase four different methods for explaining Table QA predictions: Text-to-SQL, DATER, CoTable, and **POS** (ours). Each method offers a unique approach to visualizing explanations.

- Text-to-SQL (Rajkumar et al., 2022) directly converts a natural language query into SQL, which outputs the answer. While it provides a clear, logical sequence, interpreting SQL requires expertise, limiting accessibility for non-experts (see in Fig. 4).
- DATER (Ye et al., 2023) explanations contain Subtable Selection, contextual information (i.e., the support information that was fact-checked on the input table), and attribution maps that reveal which input features influence the prediction (see in Fig. 5).
- CoTable (Wang et al., 2023) presents abstract functions, intermediate tables, and attribution maps, showing table transformations step-by-step (see in Fig. 6).

---

**Statement: the wildcats kept the opposing team scoreless in four games**

**Input Table: 1947 kentucky wildcats football team**

| game | date | opponent | result | wildcats_points | opponents | record |
|---|---|---|---|---|---|---|
| 1 | 9999-09-20 | ole miss | loss | 7 | 14 | 0 - 1 |
| 2 | 9999-09-27 | cincinnati | win | 20 | 0 | 1 - 1 |
| 3 | 9999-10-04 | xavier | win | 20 | 7 | 2 - 1 |
| 4 | 9999-10-11 | 9 georgia | win | 26 | 0 | 3 - 1 , 20 |
| 5 | 9999-10-18 | 10 vanderbilt | win | 14 | 0 | 4 - 1 , 14 |
| 6 | 9999-10-25 | michigan state | win | 7 | 6 | 5 - 1 , 13 |
| 7 | 9999-11-01 | 18 alabama | loss | 0 | 13 | 5 - 2 |
| 8 | 9999-11-08 | west virginia | win | 15 | 6 | 6 - 2 |
| 9 | 9999-11-15 | evansville | win | 36 | 0 | 7 - 2 |
| 10 | 9999-11-22 | tennessee | loss | 6 | 13 | 7 - 3 |

**SQL Command:**

```
SELECT
    CASE
        WHEN COUNT(*) = 4 THEN 'TRUE'
        ELSE 'FALSE'
    END
FROM table_sql
WHERE opponents = 0;
```

Figure 4: **Text-to-SQL** explanations provide only the SQL command, which is intuitive for domain experts.

**Statement: the wildcats kept the opposing team scoreless in four games in the table: the wildcats kept the opposing team scoreless in 4 games.**

**Input Table: 1947 kentucky wildcats football team**

Step 1: Select Rows (row 4, row 5, row 3, row 2, row 9) and Select Columns (opponents, wildcats points, game)

| game | date | opponent | result | wildcats points | opponents | record |
|---|---|---|---|---|---|---|
| 1 | sept 20 | ole miss | loss | 7 | 14 | 0 - 1 |
| 2 | sept 27 | cincinnati | win | 20 | 0 | 1 - 1 |
| 3 | oct 4 | xavier | win | 20 | 7 | 2 - 1 |
| 4 | oct 11 | 9 georgia | win | 26 | 0 | 3 - 1 , 20 |
| 5 | oct 18 | 10 vanderbilt | win | 14 | 0 | 4 - 1 , 14 |
| 6 | oct 25 | michigan state | win | 7 | 6 | 5 - 1 , 13 |
| 7 | nov 1 | 18 alabama | loss | 0 | 13 | 5 - 2 |
| 8 | nov 8 | west virginia | win | 15 | 6 | 6 - 2 |
| 9 | nov 15 | evansville | win | 36 | 0 | 7 - 2 |
| 10 | nov 22 | tennessee | loss | 6 | 13 | 7 - 3 |

Sub-table Selection

| opponents | wildcats points | game |
|---|---|---|
| 0 | 20 | 2 |
| 7 | 20 | 3 |
| 0 | 26 | 4 |
| 0 | 14 | 5 |
| 0 | 36 | 9 |

Contextual information: the wildcats kept the opposing team scoreless in 4 games.

**Prompting LLM for the final answer... >>>**

**Prediction: TRUE**

Figure 5: **DATER** explanations contain Sub-table Selection (S), contextual information (C), and highlights (H) that reveal which input features influence the prediction. In DATER, the cells used to construct the Sub-table Selection are yellow-highlighted. Additionally, the contextual information has been fact-checked using SQL commands against the input table.

**Statement: the wildcats kept the opposing team scoreless in four games**

**Input Table: 1947 kentucky wildcats football team**

Step 1: f_select_row(row 1, row 2, row 3, row 4, row 8)

| game | date | opponent | result | wildcats points | opponents | record |
|---|---|---|---|---|---|---|
| 1 | sept 20 | ole miss | loss | 7 | 14 | 0 - 1 |
| 2 | sept 27 | cincinnati | win | 20 | 0 | 1 - 1 |
| 3 | oct 4 | xavier | win | 20 | 7 | 2 - 1 |
| 4 | oct 11 | 9 georgia | win | 26 | 0 | 3 - 1 , 20 |
| 5 | oct 18 | 10 vanderbilt | win | 14 | 0 | 4 - 1 , 14 |
| 6 | oct 25 | michigan state | win | 7 | 6 | 5 - 1 , 13 |
| 7 | nov 1 | 18 alabama | loss | 0 | 13 | 5 - 2 |
| 8 | nov 8 | west virginia | win | 15 | 6 | 6 - 2 |
| 9 | nov 15 | evansville | win | 36 | 0 | 7 - 2 |
| 10 | nov 22 | tennessee | loss | 6 | 13 | 7 - 3 |

Step 2: f_select_column(game, wildcats points, opponents)

| game | date | opponent | result | wildcats points | opponents | record |
|---|---|---|---|---|---|---|
| 2 | sept 27 | cincinnati | win | 20 | 0 | 1 - 1 |
| 3 | oct 4 | xavier | win | 20 | 7 | 2 - 1 |
| 4 | oct 11 | 9 georgia | win | 26 | 0 | 3 - 1 , 20 |
| 5 | oct 18 | 10 vanderbilt | win | 14 | 0 | 4 - 1 , 14 |
| 9 | nov 15 | evansville | win | 36 | 0 | 7 - 2 |

Step 3: f_sort_column(opponents)

| game | wildcats points | opponents |
|---|---|---|
| 2 | 20 | 0 |
| 3 | 20 | 7 |
| 4 | 26 | 0 |
| 5 | 14 | 0 |
| 9 | 36 | 0 |

Step 4: simple_query()

| game | wildcats points | opponents |
|---|---|---|
| 2 | 20 | 0 |
| 4 | 26 | 0 |
| 5 | 14 | 0 |
| 9 | 36 | 0 |
| 3 | 20 | 7 |

**Prompting LLM for the final answer... >>>**

**Prediction: TRUE**

Figure 6: **CoTable** explanations present intermediate tables (T) and highlights (H), showing key steps in data transformation. In CoTable, intermediate tables and attributions (Nguyen et al., 2021) are provided. Additionally, the steps are presented through function names and their arguments.

**Statement: the wildcats kept the opposing team scoreless in four games**

**Input Table: 1947 kentucky wildcats football team**

Step 1: Order the table by 'opponents' in ascending order.

| game | date | opponent | result | wildcats_points | opponents | record |
|---|---|---|---|---|---|---|
| 1 | 9999-09-20 | ole miss | loss | 7 | 14 | 0 - 1 |
| 2 | 9999-09-27 | cincinnati | win | 20 | 0 | 1 - 1 |
| 3 | 9999-10-04 | xavier | win | 20 | 7 | 2 - 1 |
| 4 | 9999-10-11 | 9 georgia | win | 26 | 0 | 3 - 1 , 20 |
| 5 | 9999-10-18 | 10 vanderbilt | win | 14 | 0 | 4 - 1 , 14 |
| 6 | 9999-10-25 | michigan state | win | 7 | 6 | 5 - 1 , 13 |
| 7 | 9999-11-01 | 18 alabama | loss | 0 | 13 | 5 - 2 |
| 8 | 9999-11-08 | west virginia | win | 15 | 6 | 6 - 2 |
| 9 | 9999-11-15 | evansville | win | 36 | 0 | 7 - 2 |
| 10 | 9999-11-22 | tennessee | loss | 6 | 13 | 7 - 3 |

Step 2: Select rows where 'opponents' is 0.

| game | date | opponent | result | wildcats_points | opponents | record |
|---|---|---|---|---|---|---|
| 2 | 9999-09-27 | cincinnati | win | 20 | 0 | 1 - 1 |
| 4 | 9999-10-11 | 9 georgia | win | 26 | 0 | 3 - 1 , 20 |
| 5 | 9999-10-18 | 10 vanderbilt | win | 14 | 0 | 4 - 1 , 14 |
| 9 | 9999-11-15 | evansville | win | 36 | 0 | 7 - 2 |
| 6 | 9999-10-25 | michigan state | win | 7 | 6 | 5 - 1 , 13 |
| 8 | 9999-11-08 | west virginia | win | 15 | 6 | 6 - 2 |
| 3 | 9999-10-04 | xavier | win | 20 | 7 | 2 - 1 |
| 7 | 9999-11-01 | 18 alabama | loss | 0 | 13 | 5 - 2 |
| 10 | 9999-11-22 | tennessee | loss | 6 | 13 | 7 - 3 |
| 1 | 9999-09-20 | ole miss | loss | 7 | 14 | 0 - 1 |

Step 3: Use a `CASE` statement to return TRUE if the number of rows is equal to 4, otherwise return FALSE.

| game | date | opponent | result | wildcats_points | opponents | record |
|---|---|---|---|---|---|---|
| 2 | 9999-09-27 | cincinnati | win | 20 | 0 | 1 - 1 |
| 4 | 9999-10-11 | 9 georgia | win | 26 | 0 | 3 - 1 , 20 |
| 5 | 9999-10-18 | 10 vanderbilt | win | 14 | 0 | 4 - 1 , 14 |
| 9 | 9999-11-15 | evansville | win | 36 | 0 | 7 - 2 |

| verification_result |
|---|
| TRUE |

**Prediction: TRUE**

Figure 7: **POS** (ours) explanation contains input Table T, input query Q, and prediction P, intermediate Table T, highlights H. The green-highlighted cells indicate where the information in the table matches the conditions specified in the natural language steps.

# B  LLM-ONLY, PROGRAM-ONLY, & HYBRID APPROACHES FOR TABLE QA

**LLM-only.**  These approaches rely solely on LLMs to generate answers without explicitly performing table transformations. End-to-End QA prompts the LLM to generate answers directly from the input table and question. Similarly, Few-Shot QA (Brown et al., 2020) includes few-shot examples $(T, Q, A)$ as the context to aid the LLM. In contrast, Chain-of-Thought (Wei et al., 2022) prompts the LLM to explain its reasoning process step-by-step before delivering the final answer.

**Program-only.**  Program-based approaches generate explicit programs to perform table transformation and answer the question. Latent Program Algorithm (LPA) (Chen et al., 2020) frames Tab-Fact verification as a program synthesis task, converting input queries into sequential operations (e.g., min, max, count, filter) executed via Python-Pandas. On the other hand, Text-to-SQL (Rajkumar et al., 2022) translates a natural language query directly into a single SQL command, which is then applied to the input table to generate the answer.

**Hybrid.**  Hybrid approaches combine the strengths of LLM reasoning and programs to perform Table QA and achieve state-of-the-art performance. Dater (Ye et al., 2023) uses an LLM to extract relevant sub-tables, while breaking queries into sub-queries and executing SQL commands to retrieve factual information. Similarly, TabSQLify (Nahid & Rafiei, 2024) leverages LLMs to generate SQL commands, which is then used to create query-focused sub-tables. Binder (Cheng et al., 2023) takes a different approach by converting natural language questions into executable programs. It blends API calls with symbolic language interpreters like SQL or Python to address reasoning gaps that cannot be handled through offline methods alone. Lastly, CoTable (Wang et al., 2023) dynamically plans a sequence of predefined table operations–such as selecting rows or adding columns, allowing it to iteratively transform the table based on the intermediate information. Despite their differences, Dater, TabSQLify, and CoTable all share a common strategy: they input the final simplified table along with the original query into an LLM to produce the final answer.

# C  ABLATION STUDY FOR POS

To study the contributions of each component in **POS**, we perform ablation studies on TabFact and WikiTQ. Due to the deprecation of `gpt-3.5-turbo-16k-0613`[2], we use `gpt-4o-mini` for these experiments. Tab. 4 summarizes the results of our ablation studies on three key components:

Table 4: Ablation studies of **POS** components on TabFact and WikiTQ with `gpt-4o-mini`. The check-marks indicate the inclusion of a component, while crosses indicate its removal. "Fallback" refers to the percentage of samples unsolvable by **POS**, which are instead handled by fallback to end-to-end QA, as in (Chen, 2023; Kong et al., 2024).

| Component | Atomicity | Planning | Step-to-SQL | TabFact (%) | | WikiTQ (%) | |
|---|---|---|---|---|---|---|---|
| | | | | Accuracy | Fallback | Accuracy | Fallback |
| **POS** | ✓ | ✓ | ✓ | 77.22 | 11.56 | 48.90 | 22.63 |
| w/o Atomicity | ✗ | ✓ | ✓ | 78.11 | 11.85 | 50.02 | 23.29 |
| w/o NL Planning | ✓ | ✗ | ✓ | 77.90 | 40.61 | 48.27 | 42.84 |
| w/o Step-to-SQL | ✓ | ✓ | ✗ | 78.26 | 53.06 | 27.05 | 41.64 |

**Atomicity**    To assess the importance of atomic steps, we remove the constraint of atomicity in the planning step in Sec. 3.1 as well as in in-context examples (**w/o Atomicity**). This means the LLM is allowed to generate plans with more complex, compound steps. Unexpectedly, we observe an improvement in the accuracy of **POS** on both datasets. We hypothesize that while the steps become more complex, `gpt-4o-mini` is still able to handle the Step-to-SQL conversion successfully. This is evident from the minimal impact on the fallback rate. However, we argue that interpretability is affected due to the increased complexity of the plan steps, making it more challenging for users to comprehend and trust the model's reasoning process. We present qualitative examples of **POS** explanations with and without atomicity in Appendix D.

**NL Planning**    We replace the natural-language planning with a direct prompt that asks the LLM to generate a sequence of SQL commands to solve the question end-to-end (**w/o NL Planning**). We find that this component has minimal impact on the model's accuracy. However, we observe a significant increase in fallback rate–rising from 11.56% to 40.61% on TabFact and from 22.63% to 42.84% on WikiTQ. This indicates that many of the generated SQL-based plans are unexecutable due to syntax errors or logical inconsistencies (e.g., referring to non-existent columns), significantly hurting model interpretability.

**Step-to-SQL Conversion**    We modify the table transformation process to rely on prompting LLMs rather than executing SQL. Specifically, we ask the LLM to transform the table based on the natural-language steps, substituting the Step-to-SQL conversion with black-box LLM reasoning (**w/o Step-to-SQL**). This leads to a negligible increase in accuracy on TabFact but a substantial drop on WikiTQ (from 48.90% to 27.05%), indicating that relying on the LLM for table transformations can severely impact model accuracy. We argue that this is likely due to the LLM's likelihood for hallucinations or errors when handling complex tables (Chen, 2023; Wang et al., 2023). Additionally, this approach diminishes interpretability, as the table transformations are no longer transparent or traceable.

---

[2]https://platform.openai.com/docs/deprecations/2023-11-06-chat-model-updates

# D QUALITATIVE EXAMPLES FOR POS EXPLANATIONS WITHOUT ATOMICITY

Below, we provide qualitative examples of POS explanations with and without atomicity in NL Planning. Removing atomicity from the plan steps can negatively impact interpretability, as the added complexity makes it harder for users to understand and trust the model's reasoning process.

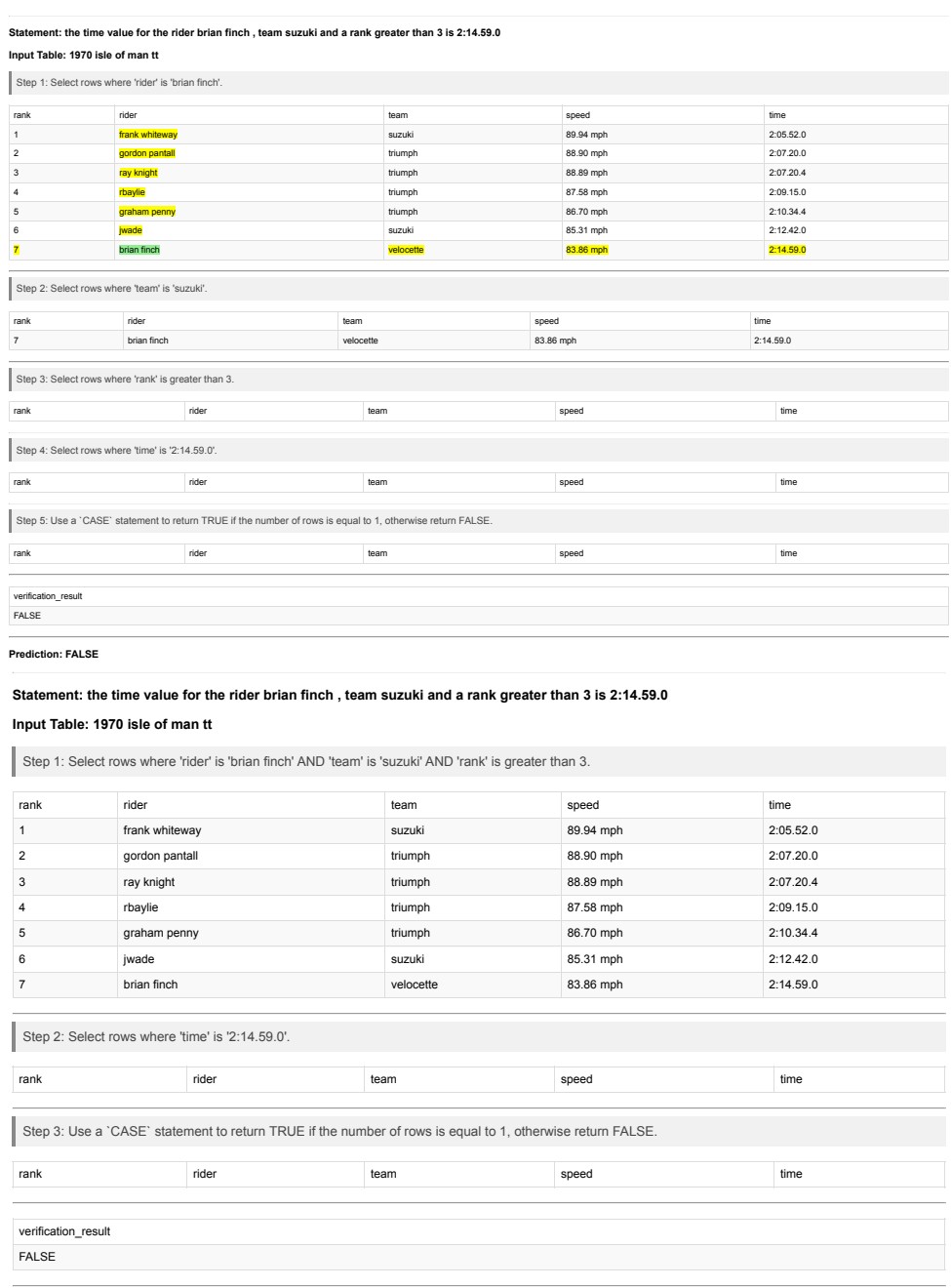

Figure 8: Upper: POS explanation with atomicity. Lower: POS explanation without atomicity.

**Statement: after 1985 , the united states contributed two players with rafael araãjo being the most recent**

**Input Table: utah jazz all - time roster**

Step 1: Select rows where 'nationality' is 'united states' and 'years_for_jazz' is after 1985.

| player | nationality | position | years_for_jazz | school___club_team |
|---|---|---|---|---|
| rick adelman | united states | guard | 1974-01-01 | loyola (ca) |
| john amaechi | england | center / forward | 2001-03-01 | penn state |
| louis amundson | united states | forward | 2007-01-01 | unlv |
| j j anderson | united states | forward | 1982-01-01 | bradley |
| shandon anderson | united states | guard / forward | 9999-01-01 | georgia |
| rafael araãjo | brazil | center | 2006-01-01 | byu |
| carlos arroyo | puerto rico | guard | 2002-05-01 | florida international |
| isaac austin | united states | center | 1991-01-01 | arizona state |
| anthony avent | united states | forward | 1998-01-01 | seton hall |

Step 2: Use a `CASE` statement to return TRUE if the number of rows is equal to 2, otherwise return FALSE.

| player | nationality | position | years_for_jazz | school___club_team |
|---|---|---|---|---|
| louis amundson | united states | forward | 2007-01-01 | unlv |
| shandon anderson | united states | guard / forward | 9999-01-01 | georgia |
| isaac austin | united states | center | 1991-01-01 | arizona state |
| anthony avent | united states | forward | 1998-01-01 | seton hall |

| verification_result |
|---|
| FALSE |

**Prediction: FALSE**

**Statement: galina voskoboeva played a total of 3 games on a hard tennis court , and 1 on clay**

**Input Table: galina voskoboeva**

Step 1: Select rows where 'surface' is 'hard' and count the number of such rows to determine the total games played on hard courts.

| outcome | date | tournament | surface | opponent | score |
|---|---|---|---|---|---|
| runner - up | 2003-01-28 | tipton | hard (i) | matea mezak | 6 - 4 , 4 - 6 , 4 - 6 |
| winner | 2003-06-29 | mont - de - marsan | hard (i) | oleksandra kravets | 6 - 4 , 6 - 2 |
| runner - up | 2003-10-03 | latina | clay | roberta vinci | 3 - 6 , 4 - 6 |
| runner - up | 2005-11-08 | pittsburgh | hard | lilia osterloh | 6 - 7 , 4 - 6 |
| winner | 2006-06-06 | cuneo , italy | clay | alice canepa | 6 - 1 , 6 - 2 |

Step 2: Select rows where 'surface' is 'clay' and count the number of such rows to determine the total games played on clay courts.

| outcome | date | tournament | surface | opponent | score | hard_court_games |
|---|---|---|---|---|---|---|
| runner - up | 2005-11-08 | pittsburgh | hard | lilia osterloh | 6 - 7 , 4 - 6 | 1 |

Step 3: Use a CASE statement to return TRUE if the count of hard court games is equal to 3 AND the count of clay court games is equal to 1, otherwise return FALSE.

| total_clay_games |
|---|
| 0 |

| verification_result |
|---|
| FALSE |

**Prediction: FALSE**

Figure 9: Two **POS** explanations without atomicity. The steps are compound and the attribution maps are non-trivial to comprehend.

# E   EXTRACTING COLUMNS FROM SQL COMMANDS

We design an algorithm to analyze SQL queries and identify the columns used within them.

## E.1   ALGORITHM OVERVIEW

The algorithm follows these main steps:

1. **Preprocessing:** Remove comments and normalize whitespace in the SQL query.
2. **Column Extraction:** Parse different clauses of the SQL query to identify column names:
   - SELECT clause: Extract both regular columns and those used in functions.
   - WHERE clause: Identify columns used in conditions.
   - ORDER BY clause: Extract columns used for sorting.
3. **Filtering:** Compare extracted columns against a list of original columns to ensure validity.

## E.2   IMPLEMENTATION DETAILS

The algorithm is implemented using regular expressions to parse the SQL query. Key implementation details include:

- Use of re.sub() for comment removal and whitespace normalization.
- Application of re.search() and re.findall() for extracting column names from different parts of the query.
- Special treatment for columns used within functions in the SELECT, WHERE, ORDER BY clauses.

## E.3   AN EXAMPLE OF DATA-ATTRIBUTION TRACKING FOR TABLE QA

Here, we use the table transformation in Fig. 2– **3** as an example to illustrate our data-attribution tracking algorithm (Fig. 10):

- **(a)** Adding the Tracking Index Column
- **(b)** Executing the SQL Command
- **(c)** Identifying Selected Rows
- **(d)** Parsing SQL Commands to Identify Selected Columns
- **(e)** Mapping to Original Indices

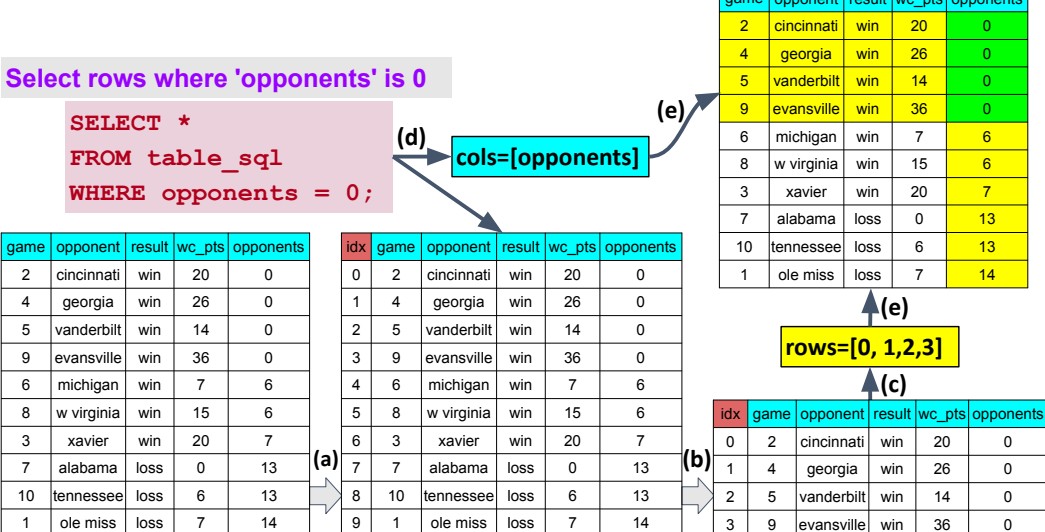

Figure 10: Data-attribution tracking algorithm.

## F HALLUCINATIONS IN SUB-TABLE SELECTION

Methods like CoTable and DATER aim to answer questions by performing complex table transformations—specifically, selecting sub-tables from the input table based on reasoning steps. However, these methods are prone to errors regarding which table entries to select, leading to irrational or irrelevant information being considered in the final answer.

As illustrated in Fig. 11, although Chain-of-Table correctly answers the question *Q: True or False? In four different baseball games, the final score was 9-2*, it irrationally selects unrelated information (game 3) from the input table. Similarly, DATER, shown in Fig. 12, selects rows 2, 3, 4, 5, and 9 to answer the same question. However, the inclusion of row 3 is illogical and does not contribute to a valid answer.

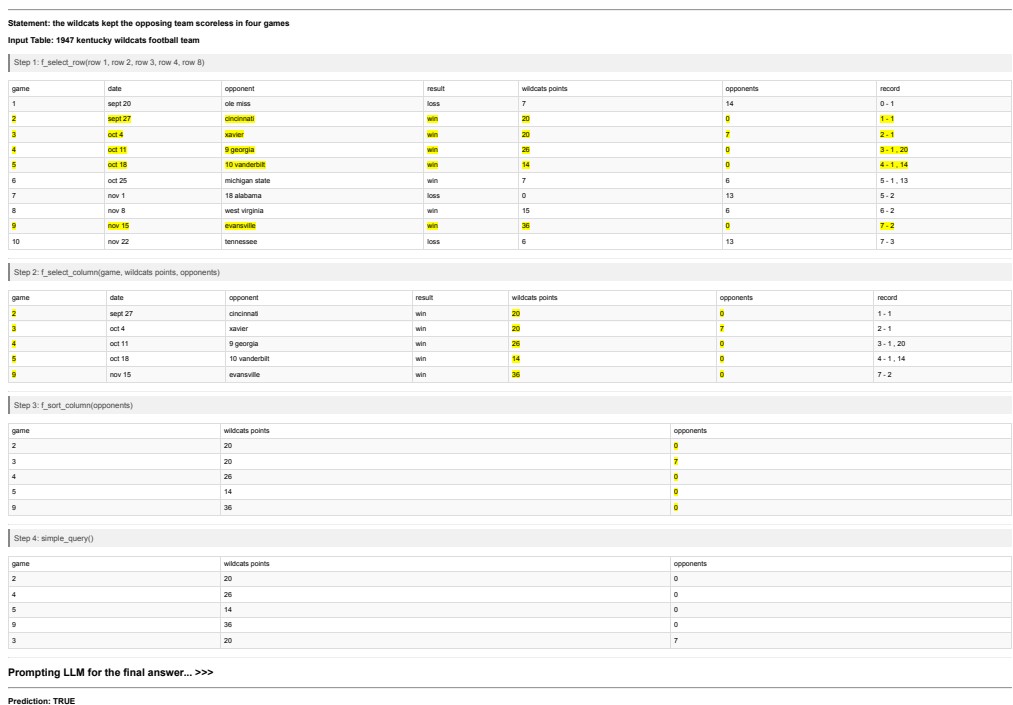

Figure 11: Although CoTable correctly answers the question *Q: True or False? In four different baseball games, the final score was 9-2*, it irrationally selects unrelated information (game 3) from the input table.

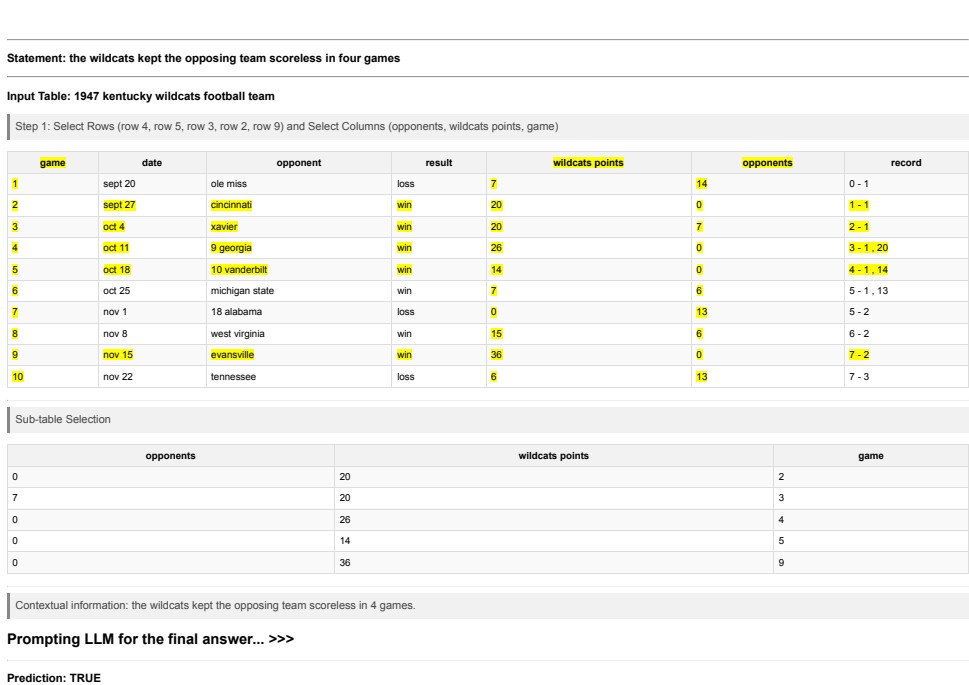

**Statement: the wildcats kept the opposing team scoreless in four games**

**Input Table: 1947 kentucky wildcats football team**

Step 1: Select Rows (row 4, row 5, row 3, row 2, row 9) and Select Columns (opponents, wildcats points, game)

| game | date | opponent | result | wildcats points | opponents | record |
|------|------|----------|--------|-----------------|-----------|--------|
| 1 | sept 20 | ole miss | loss | 7 | 14 | 0 - 1 |
| 2 | sept 27 | cincinnati | win | 20 | 0 | 1 - 1 |
| 3 | oct 4 | xavier | win | 20 | 7 | 2 - 1 |
| 4 | oct 11 | 9 georgia | win | 26 | 0 | 3 - 1 , 20 |
| 5 | oct 18 | 10 vanderbilt | win | 14 | 0 | 4 - 1 , 14 |
| 6 | oct 25 | michigan state | win | 7 | 6 | 5 - 1 , 13 |
| 7 | nov 1 | 18 alabama | loss | 0 | 13 | 5 - 2 |
| 8 | nov 8 | west virginia | win | 15 | 6 | 6 - 2 |
| 9 | nov 15 | evansville | win | 36 | 0 | 7 - 2 |
| 10 | nov 22 | tennessee | loss | 6 | 13 | 7 - 3 |

Sub-table Selection

| opponents | wildcats points | game |
|-----------|-----------------|------|
| 0 | 20 | 2 |
| 7 | 20 | 3 |
| 0 | 26 | 4 |
| 0 | 14 | 5 |
| 0 | 36 | 9 |

Contextual information: the wildcats kept the opposing team scoreless in 4 games.

**Prompting LLM for the final answer... >>>**

**Prediction: TRUE**

Figure 12: DATER selects rows 2, 3, 4, 5, and 9 to answer the question. However, the inclusion of row 3 is illogical and does not contribute to a valid answer.

# G DETAILS FOR LLM-AS-A-JUDGE EXPERIMENTS

## G.1 PROMPT FOR LLM-AS-A-JUDGE IN FORWARD SIMULATION

---

**Prompt for LLM-as-a-Judge in Forward Simulation**

prompt = f""" Given an input statement, an Artificial Intelligence (AI) model will output either TRUE or FALSE. Your job in this Simulation task is to use the AI's explanation to guess the machine response. Specifically, please choose which response (TRUE/FALSE) model would output regardless of whether you think that response is correct or not.

Explanation: [**text_content**]

Based on this explanation, guess what the model will predict on the Statement based on the provided explanation. Answer with only 'TRUE' or 'FALSE': """

---

## G.2 PROMPT FOR LLM-AS-A-JUDGE IN PREFERENCE RANKING

It is well known that LLM-as-a-Judge exhibits a strong bias toward the position of the options presented to it (Dubois et al., 2024). To eliminate this bias in our prompt, we shuffle the order of the four methods four times and compute the average ranking.

---

**Prompt for LLM-as-a-Judge in Preference Ranking**

```
    prompts = []

    num_methods = len(methods)
```
# Create a dictionary mapping methods to their descriptions

method_descriptions = {

"DATER": """DATER is a method that focuses on selecting relevant information from the input table and providing contextual information to support the statement verification process. The explanation contains:

1. Sub-table Selection: Dater selects a sub-table from the original input Table that is relevant to the Statement.

2. Contextual Information: Dater provides contextual information that is fact-checked against the Table.""",

"COT": """COT is a method that breaks down the question-answering process into a series of intermediate tables. Each step in the chain represents a specific operation on the table, leading to the final answer. The explanation contains:

1. Step Descriptions: Each step is accompanied by a function with arguments, providing context for the transformation.

2. Intermediate Tables: We display the intermediate tables resulting from each function, showing the state of the data at each step.

3. Row and Column Highlighting: Rows and Columns used in the current step are highlighted with background-color:yellow.""",

"Text2SQL": """Text2SQL is a method that translates the natural language query into a single SQL query. The SQL query itself serves as the explanation for how the system arrives at its answer. The explanation contains: The generated SQL command that will be directly applied onto the table to generate the final answer.""",

"POS": """POS is a Table QA method that breaks down the question-answering process into a series of natural-language steps. Each step represents a specific operation on the table, leading to the final answer. The explanation contains:

1. Step Descriptions: Each step is accompanied by a natural-language description of the atomic step performed, providing context for the transformation.

2. Intermediate Tables: We display the intermediate tables resulting from each step, showing the state of the data at each step.

3. Attribution Maps: We highlight the the rows, columns, and cells involved in each table transformation over intermediate tables. Row and Column Highlighting: Rows and Columns used in the current step are highlighted with background-color:yellow. Cell Highlighting: Cells that directly match the conditions in the current step are highlighted with background-color:90EE90.""" }

```
    for i in range(num_methods):
        shuffled_methods = methods[i:] + methods[:i]
```

---

```
prompt = f""" You are given explanations from four different methods for the same table fact verification task.
Please rank these explanations based on their clarity, coherence, and helpfulness in understanding the model's
reasoning.

Clarity Definition: How easy is the explanation to understand? Is the language clear and straightforward?

Coherence Definition: Does the explanation logically flow and make sense as a whole? Are the ideas well-
connected?

Helpfulness in Understanding the Model's Reasoning Definition: How effectively does the explanation help you
understand why the model made its decision? Does it reveal the reasoning process?

Provide the ranking from best to worst.

Explanations:
"""
```

# H PROMPT ENGINEERING

## H.1 PROMPT FOR ATOMIC PLANNING FOR TABFACT

### H.1.1 DECOMPOSITION OF QUERY $Q$

The decomposition process breaks down the complex query $Q$ into a sequence of atomic steps. This is achieved through a carefully crafted prompt provided to the LLM. The prompt includes:

- **Instructional Guidelines:** We instruct the LLM to "Develop a step-by-step plan to answer the question given the input table".
- **Emphasis on Atomicity:** The LLM is instructed that "Each step in your plan should be very atomic and straightforward, ensuring they can be easily executed or converted into SQL".
- **In-context Examples:** We provide example inputs $(T, Q)$ along with their corresponding plans to serve as in-context examples for planning (see Appendix I).

### H.1.2 SEQUENCING OF STEPS

Correct sequencing is crucial because each step depends on the output of the previous one. We ensure proper sequencing by:

- **Explicit Instructions:** The LLM is instructed that "The order of steps is crucial! You must ensure the orders support the correct information retrieval and verification!".
- **Dependencies:** Clarifying that "The next step will be executed on the output table of the previous step. The first step will be executed on the given Table".
- **Handling Comparatives and Superlatives:** Instructing the LLM on how to handle statements involving terms like 'highest', 'lowest', etc., by ordering the table before selecting rows.

---

**Prompt for atomic planning**

[**In-context examples**]

**### Here come to your task!**

**Table caption:** {caption}

/ ∗ {table2string(table_info["table_text"])} ∗ / # Convert Table into markdown format

**This Table has {num_rows} rows.**

**Statement:** {sample["statement"]}

Let's develop a step-by-step plan to verify if the given **Statement** is **TRUE** or **FALSE** on the given **Table**!

You **MUST** think carefully analyze the **Statement** and comprehend it before writing the plan!

**Plan Steps:** Each step in your plan should be very atomic and straightforward, ensuring they can be easily executed or converted into SQL.

You **MUST** make sure all conditions (except those mentioned in the table caption) are checked properly in the steps.

**Step order:** The order of steps is crucial! You must ensure the orders support the correct information retrieval and verification!

The next step will be executed on the output table of the previous step. The **first step** will be executed on the given **Table**.

For comparative or superlative **Statement** involving "highest," "lowest," "earliest," "latest," "better," "faster," "earlier," etc., you should order the table accordingly before selecting rows. This ensures that the desired comparative or superlative data is correctly retrieved.

**Plan:**

---

### H.1.3 THE IMPORTANCE OF STEP ORDER

In this example, step 1 is crucial. If the table is not ordered by 'rank' first, selecting row number 1 (step 2) or filtering by 'athlete' (step 3) will return the wrong result. Only by ensuring that the table is correctly ordered beforehand can we reliably select the top-ranked athlete. Thus, the sequence of steps must be followed precisely to avoid logical errors.

**A plan where the step order determines the correctness**

**Table:** Olympic 2018; Table Tennis

```
/*
col : rank| athlete             | time
row 1 : 1 | manjeet kaur (ind)   | 52.17
row 2 : 2 | olga tereshkova (kaz) | 51.86
row 3 : 3 | pinki pramanik (ind)  | 53.06
*/
```

**Statement:** manjeet had the highest rank in the competition.

**Plan:**

1. Order the table by 'rank' in ascending order.
2. Select row number 1.
3. Select rows where 'athlete' is 'manjeet' using the `LIKE` function.
4. Use a `CASE` statement to return TRUE if the number of rows is equal to 1, otherwise return FALSE.

## H.2 PROMPT FOR STEP-TO-SQL

**Prompt for Step-to-SQL**

[**In-context examples**]

Given this table:

`/* {table2string(intermediate_table)} */`

Data types of columns:

- {col_1}: {dtype_str_1}
- {col_2}: {dtype_str_2}
- …

Write a SQL command that: {natural_language_step}

The original table has {num_rows} rows.

**Constraints for your SQL:**

1. If using SELECT COUNT(*), SUM, MAX, AVG, you MUST use AS to name the new column. If adding new columns, they should be different than columns {existing_cols}.
2. Your SQL command **MUST** be compatible and executable by Python `sqlite3` and `pandas`.
3. If using `FROM`, the table to be selected **MUST** be {table_name}.

# I    IN-CONTEXT EXAMPLES

## I.1    IN-CONTEXT EXAMPLES FOR ATOMIC PLANNING

---

**In-context examples for atomic planning**

**TabFact**

**Table:** 2005 tournament results

```
/*
col  : id | name    | hometown     | score
row 1 : 1 | alice   | new york     | 85
row 2 : 2 | bob     | los angeles  | 90
row 3 : 3 | charlie | chicago      | 75
row 4 : 4 | dave    | new york     | 88
row 5 : 5 | eve     | los angeles  | 92
*/
```

**Statement:** in 2005 tournament, bob and charlie are both from chicago.

**Plan:** # Natural-language step

1. Select rows where the 'name' is 'bob' or 'charlie'.
2. Select rows where 'hometown' is 'chicago'.
3. Use a `CASE` statement to return TRUE if the number of rows is equal to 2, otherwise return FALSE.

**WikiTQ**

**Table:** 2005 tournament results

```
/*
col  : id | name    | hometown     | score
row 1 : 1 | alice   | new york     | 85
row 2 : 2 | bob     | los angeles  | 90
row 3 : 3 | charlie | chicago      | 75
*/
```

**Question:** which players are from chicago?

**Plan:** # Natural-language step

1. Select rows where the 'hometown' is 'chicago'.
2. Select the 'name' column.

---

## I.2    IN-CONTEXT EXAMPLES FOR STEP-TO-SQL

---

**In-context examples for Step-to-SQL**

Given this table:

```
/*
col : id  | name    | department | salary | years
row 1 : 1 | alice   | it         | 95000  | 3
row 2 : 2 | bob     | finance    | 105000 | 5
row 3 : 3 | charlie | marketing  | 88000  | 2
*/
```

Write a SQL command that: Select rows where the 'salary' is greater than 100000.

SQL is:

```
SELECT *
FROM table_sql
WHERE salary > 100000;
-- Select rows where the 'salary' is greater than 100000.
```

---

## J ERROR ANALYSIS OF **POS**

We notice that most errors in **POS** come from the planning stage rather than the Step-to-SQL process. The common issues are due to missing condition checks (see Fig. 13, Fig. 15, Fig. 14, Fig. 16, Fig. 17) in atomic steps. An interesting error due to the exact-matching nature of SQL can also be found in Fig. 18.

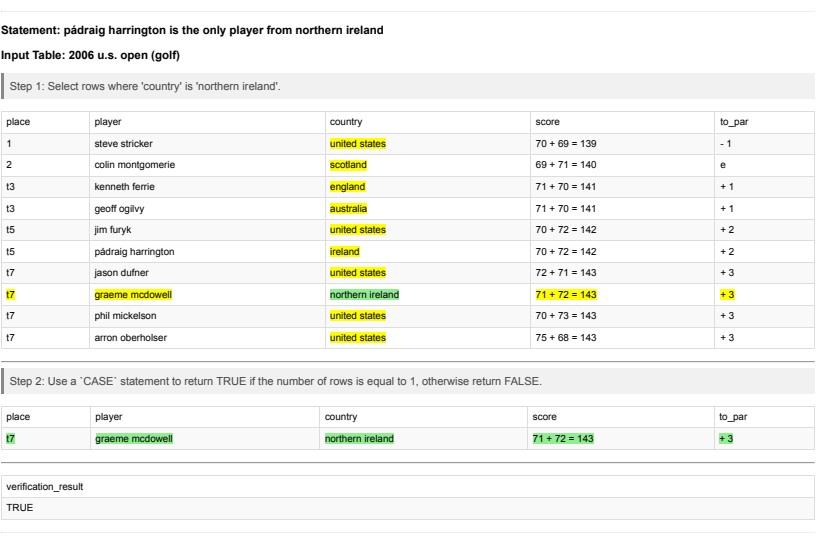

Figure 13: **POS** (ours) predicts TRUE but the groundtruth is FALSE (False Positive). In planning, **POS** misses checking the player name.

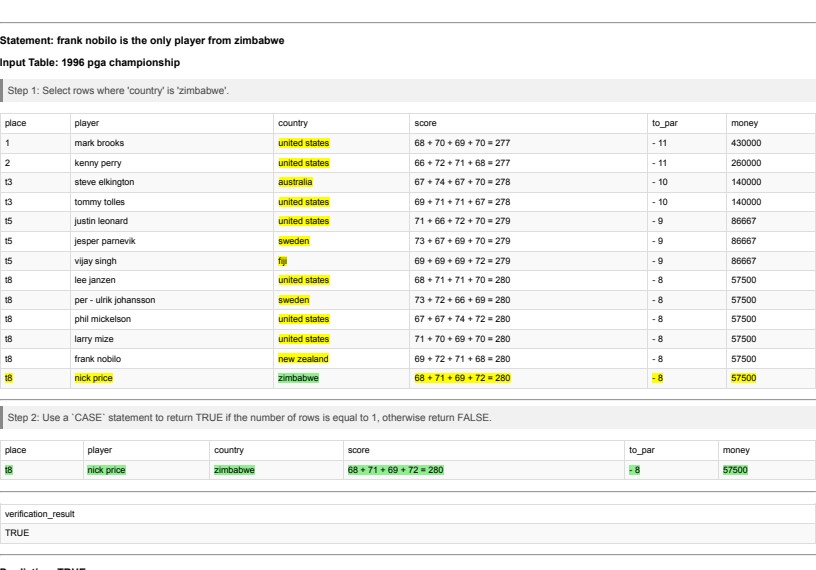

Figure 14: **POS** (ours) predicts TRUE but the groundtruth is FALSE (False Positive). In planning, **POS** misses checking the player name.

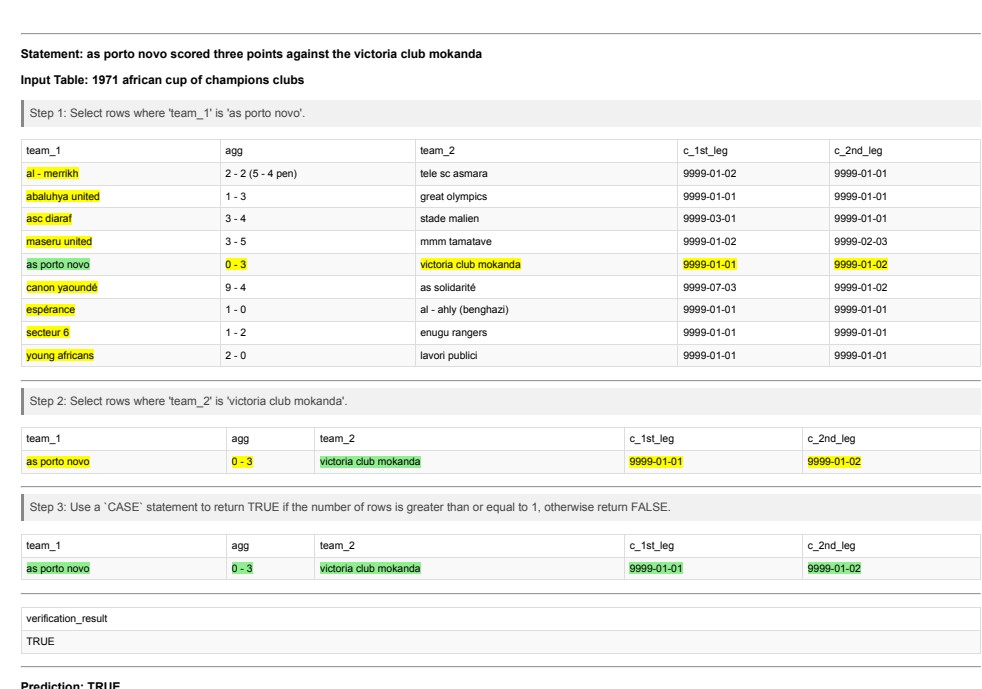

Figure 15: **POS** (ours) predicts TRUE but the groundtruth is FALSE (False Positive). In planning, **POS** misses checking the score.

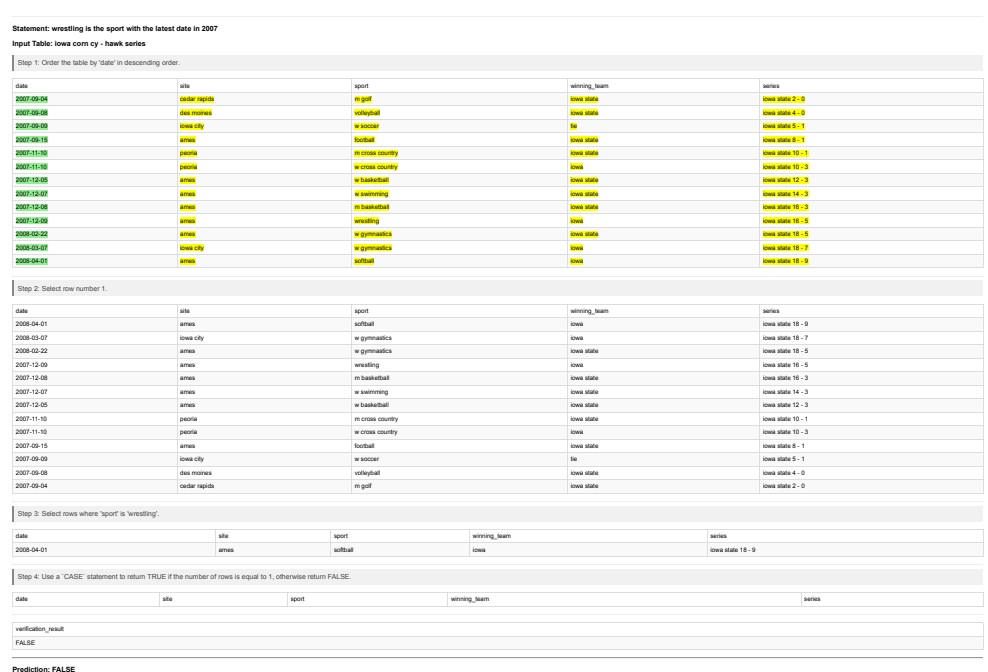

Figure 16: **POS** (ours) predicts FALSE but the groundtruth is TRUE (False Negative). In planning, **POS** misses checking the year.

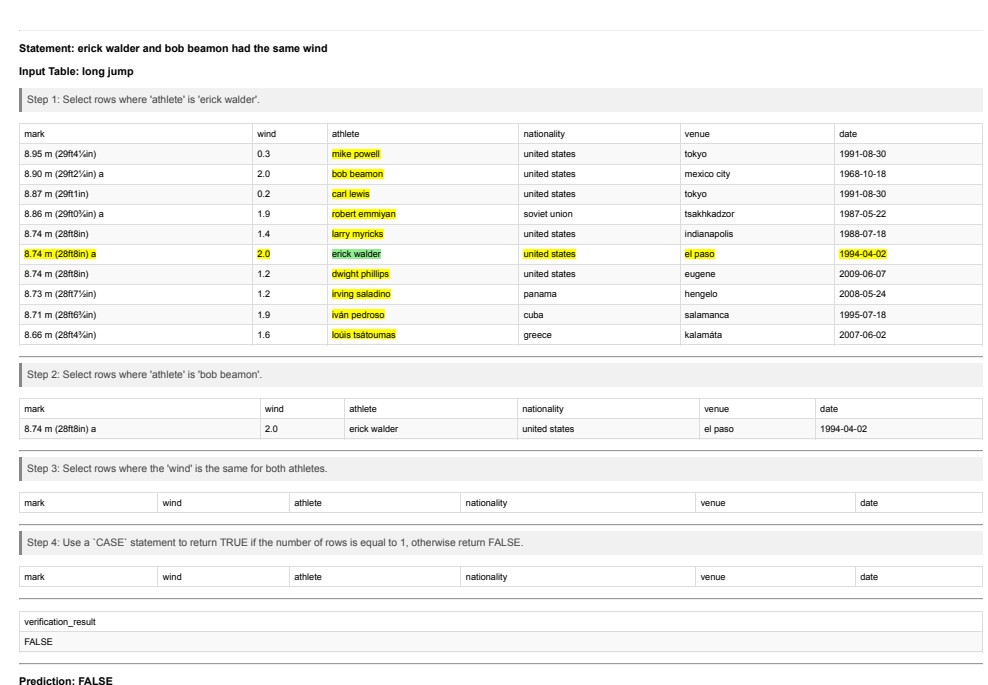

Figure 17: **POS** (ours) predicts FALSE but the groundtruth is TRUE (False Negative). In planning, **POS** should select two rows at the same step.

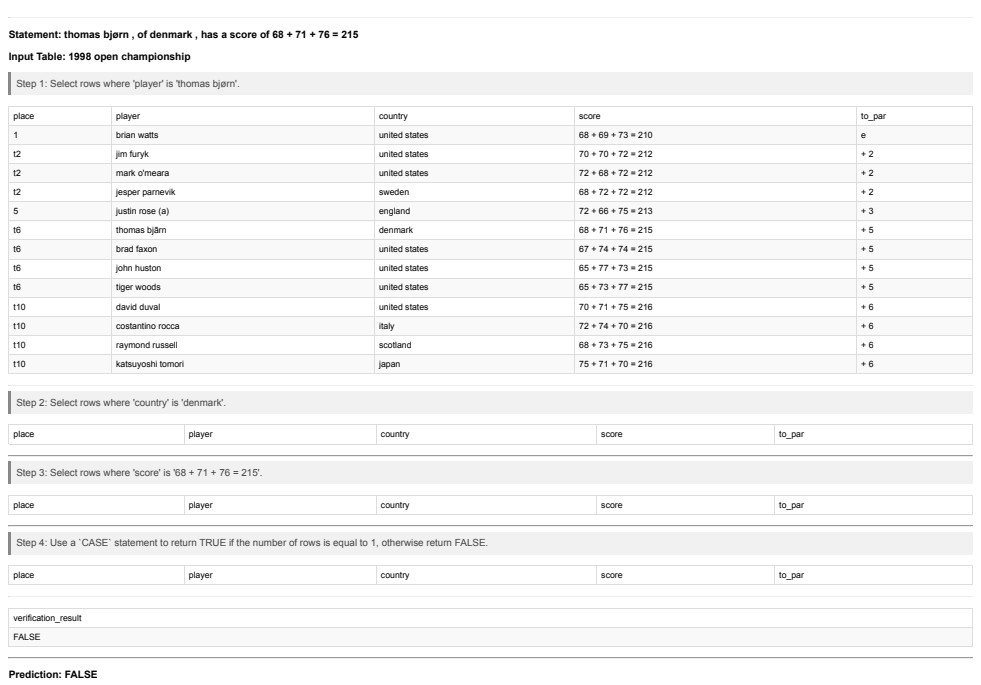

Figure 18: **POS** (ours) predicts FALSE but the groundtruth is TRUE (False Negative). The exact-matching nature of SQL makes **POS** cannot retrieve the relevant information.

## K    HUMAN STUDY INTERFACE

To evaluate the effectiveness of different explanation methods in Table QA on human users, we develop a web-based interface using HuggingFace Gradio and Flask. The interface is designed for Forward Simulation to guide participants through the study seamlessly, ensuring they understand the tasks and provide valuable feedback.

Overview of the Forward Simulation Interface Flow:

1. Informed Consent $\Rightarrow$
2. Introduction to Table QA and Forward Simulation $\Rightarrow$
3. Introduction to Explanations in Table QA $\Rightarrow$
4. Welcome page where users are asked to choose one of 4 XAI methods $\Rightarrow$
5. Specific explanation page for the chosen method $\Rightarrow$
6. Experiment pages for 10 samples $\Rightarrow$
7. Completion page!

## K.1 INFORMED CONSENT

---

# Informed Consent for Table QA Study

**Study Information:**
You are invited to participate in a research study on Table QA systems. This study aims to improve how AI systems explain their reasoning when answering questions based on tabular data.

**Study Duration and Requirements:**
1. The entire study will take approximately **20 minutes** to complete.

2. Please perform this study on a **computer** (not a phone).

3. Do not seek help from the Internet or other people during the study.

**Study Structure:**
1. Introduction to Table QA and task explanation

2. Main study: 10 questions about Table QA explanations

**Benefits:**
Your participation will contribute to the development of AI systems that can better explain their reasoning to humans, particularly in the domain of question answering from tabular data. There are no known risks associated with this study.

**Data Usage and Confidentiality:**
All data collected will be anonymized and used solely for research purposes. Your personal information will be kept confidential.

**Voluntary Participation:**
Your participation in this study is entirely voluntary. You may choose to withdraw at any time without any consequences.

**Contact Information:**
If you have any questions or concerns about this study, please contact [anonymized].

**Agreement:**
By clicking "**I Agree**" below, you confirm that you have read and understood this informed consent, and you agree to participate in this Table QA study under the terms described above.

### I Agree

---

K.2    INTRODUCTION TO TABLE QA AND FORWARD SIMULATION

# Introduction to Table QA

In this experiment, you will interact with Table QA models. Table QA involves answering questions based on data provided in tables.

## Verify if the following Statement is TRUE or FALSE

**Statement:** The Wildcats kept the opposing team scoreless in four games.
**Input Table Caption:** 1947 Kentucky Wildcats Football Team

| Game | Date | Opponent | Result | Wildcats Points | Opponents | Record |
|------|------|----------|--------|-----------------|-----------|--------|
| 1 | 9999-09-20 | Ole Miss | Loss | 7 | 14 | 0 - 1 |
| 2 | 9999-09-27 | Cincinnati | Win | 20 | 0 | 1 - 1 |
| 3 | 9999-10-04 | Xavier | Win | 20 | 7 | 2 - 1 |
| 4 | 9999-10-11 | 9 Georgia | Win | 26 | 0 | 3 - 1, 20 |
| 5 | 9999-10-18 | 10 Vanderbilt | Win | 14 | 0 | 4 - 1, 14 |
| 6 | 9999-10-25 | Michigan State | Win | 7 | 6 | 5 - 1, 13 |
| 7 | 9999-11-01 | 18 Alabama | Loss | 0 | 13 | 5 - 2 |
| 8 | 9999-11-08 | West Virginia | Win | 15 | 6 | 6 - 2 |
| 9 | 9999-11-15 | Evansville | Win | 36 | 0 | 7 - 2 |
| 10 | 9999-11-22 | Tennessee | Loss | 6 | 13 | 7 - 3 |

**Model thinks this Statement is:** TRUE

## Model Simulation Task

Given an input statement, an Artificial Intelligence (AI) model will output either TRUE or FALSE. **Your job in this Simulation task is to use the AI's explanation to guess the machine response.** Specifically, please choose which response (Statement is TRUE/Statement is FALSE) the model would output regardless of whether you think that response is correct or not.

Next

### K.3    INTRODUCTION TO EXPLANATIONS IN TABLE QA

---

# Understanding Attribution Explanations

Attribution explanations highlight specific parts of a table—such as rows, columns, or cells—that are most relevant to the answer predicted by a Table QA model. These explanations help you understand which information of the input the system considered important when predicting the answer.

**Table caption:** 1947 Kentucky Wildcats Football Team

**Statement to verify:** "The Wildcats kept the opposing team scoreless in 4 games."

| Game | Date | Opponent | Result | Wildcats Points | Opponents | Record |
|------|------|----------|--------|-----------------|-----------|--------|
| 1 | 9999-09-20 | Ole Miss | Loss | 7 | 14 | 0 - 1 |
| 2 | 9999-09-27 | Cincinnati | Win | 20 | 0 | 1 - 1 |
| 4 | 9999-10-11 | 9 Georgia | Win | 26 | 0 | 3 - 1 , 20 |
| 5 | 9999-10-18 | 10 Vanderbilt | Win | 14 | 0 | 4 - 1 , 14 |
| 9 | 9999-11-15 | Evansville | Win | 36 | 0 | 7 - 2 |

In this example, the Table QA model has highlighted specific rows and cells to explain its prediction:
1. The entire rows for games 2, 4, 5, and 9 are highlighted in yellow.

2. Within these rows, the **Opponents** column cells containing "0" or "scoreless" are highlighted in green.

These highlights indicate that the system identified four games where the opposing team did not score, verifying the statement as TRUE. The yellow highlighting shows the relevant rows, while the green highlighting represents the cells containing fine-grained information needed to verify the statement.

By using different colors for highlighting, the system provides a more nuanced explanation:
1. **Yellow highlights (rows):** Show the overall context of the relevant games.

2. **Green highlights (cells):** Pinpoint the exact information (opposing team's score of 0) that directly answer the question.

During the experiment, you will use explanations to choose which response (Statement is TRUE/ Statement is FALSE) the model would output, regardless of whether you think that response is correct or not.

## Proceed to Experiment

## K.4 WELCOME PAGE

# Let's Get Started!

**Task Instructions**

- 👤 **Enter your name**
- **#** **Choose a lucky number**
- 📊 **Select an explanation method**
- ◎ **Complete 10 samples in the experiment**

**Hi there! What is your name?**

_______________________________________________

**What is your lucky number?**

_______________________________________________

## Explanation Methods

| Chain-of-Table | Plan-of-SQLs |
|:---:|:---:|

| Text-to-SQL | Dater |
|:---:|:---:|

**Next**

## K.5   EXPERIMENT PAGE

# Sample: 1 / 10

**Please note that in select row function, starting index is 0 for Chain-of-Table and 1 for Dater and Index \* represents the selection for all rows.**

**Based on the explanation below, please guess what the AI model will predict on the input Statement below.**

# [Input Statement]

[Explanation content]

**Guess what the model will predict on the Statement based on the provided explanation?**

**Model will predict: Statement is TRUE**

**Model will predict: Statement is FALSE**

## K.6   COMPLETION PAGE

# Thank you!

You've successfully completed the experiment. Your predictions have been recorded.

**Your Labeling Accuracy**
**user_accuracy %**

**You Predicted TRUE**
**true_percentage %**

**You Predicted FALSE**
**false_percentage %**

Back to Start Page

