# OpenReview forum: "Interpretable Table Question Answering via Plans of Atomic Table Transformations"
_ICLR.cc/2025/Conference — ICLR 2025 Conference Withdrawn Submission_

### Official Review · Reviewer_8Hmg · 2024-10-29

**Soundness:** 3
**Presentation:** 2
**Contribution:** 2
**Rating:** 3
**Confidence:** 5

**Summary:**

This paper introduces Plan-of-SQLs (POS), a method for interpretable Table Question Answering that decomposes queries into simple, sequential SQL steps, ensuring full transparency in the reasoning process. POS enables users to trace how answers are derived, significantly improving interpretability over existing models. Through human and LLM evaluations, POS demonstrates strong clarity, coherence, and competitive accuracy on benchmarks like TabFact and WikiTQ, making it suitable for high-stakes applications demanding clear model explanations.

**Strengths:**

1. The paper is well-motivated, addressing the critical need for interpretability in Table QA, especially for high-stakes fields where transparency is essential.
2. It includes a comprehensive evaluation, using both human and LLM judges to assess interpretability and predictive accuracy, showcasing POS’s advantages in clarity, coherence, and overall effectiveness compared to existing methods.

**Weaknesses:**

1. **Limited Novelty**: While the paper addresses an important problem in Table QA, the core ideas, such as breaking down queries into atomic steps and using SQL transformations for interpretability, are established in other domains. This reduces the method's originality, as it largely adapts existing decomposition and programmatic interpretability approaches rather than introducing fundamentally new techniques specific to Table QA.

2. **Limited Effectiveness and Scalability**: The performance improvements with POS are incremental, indicating only a modest boost in interpretability and accuracy over baseline methods. Additionally, the approach relies on SQL-based transformations, which may restrict its applicability in more dynamic or complex data environments where SQL alone may be insufficient. As a result, the method lacks clear potential for further application in other QA scenarios or for handling more complex reasoning tasks, limiting its scalability and broader impact.

**Questions:**

See weakness

---

> ### Author Response · Authors · 2024-11-15
> **Thank You for Your Review Efforts!**
>
> Dear Reviewer `8Hmg`,
>
> We sincerely thank you for the review efforts and your thoughtful feedback on our paper. We deeply appreciate that you recognize the strengths of our work, including the clear motivation for interpretability in Table QA and your acknowledgment of our comprehensive evaluation, which demonstrates POS’s advantages in clarity, coherence, and overall effectiveness compared to existing methods.
>
> For your concerns about the technical contributions and scalability of POS, please find our detailed replies below. We hope these responses address your feedback and clarify the unique contributions and potential of POS!

---

> > ### Author Response · Authors · 2024-11-15
> > **Regarding scalability of POS**
> >
> > > Additionally, the approach relies
> > > on SQL-based transformations, which may restrict its applicability in
> > > more dynamic or complex data environments where SQL alone may be
> > > insufficient. As a result, the method lacks clear potential for further
> > > application in other QA scenarios or for handling more complex reasoning
> > > tasks, limiting its scalability and broader impact.
> >
> > Regarding the scalability concern, we recognize that tables vary significantly in format,
> > which can sometimes make SQL-based processing challenging [1].
> > However, by preprocessing tables into a SQL-compatible format (i.e., making them "SQLifiable"),
> > we can achieve significantly better interpretable Table QA.
> > For instance, the NormTab paper [2] demonstrates effective data cleaning techniques to
> > prepare tables for SQL processing, showing that, after cleaning, applying naive methods
> > like Text-to-SQL can boost accuracy on datasets like TabFact and WikiTQ by 5-10 points.
> > This shows that POS can not only provide great interpretability but also accurate answers.
> >
> > We agree that there are cases where answers cannot be generated by SQL queries alone, particularly for tasks
> > requiring long-form generation, such as text summarization tasks in datasets like QTSumm [3], which often rely on model
> > “creativity” to produce answers. For such datasets, relying solely on SQL is not ideal. We argue that combining the structured
> > data handling strengths of SQL with the creative reasoning capabilities of LLMs offers a more powerful approach.
> >
> > Finally, we want to reiterate that our primary focus in this work is on interpretability, which is why we
> > emphasize SQL-based methods. For tasks like long-form generation, hybrid explanation methods (e.g., SQL + natural-language
> > explanations) may be more suitable, and we leave this exploration for future work. We hope that POS serves as a foundation
> > for further interpretability research in Table QA.
> >
> > References:
> > - [1] On the potential of lexico-logical alignments for semantic parsing to sql queries.
> > - [2] NormTab: Improving Symbolic Reasoning in LLMs Through Tabular Data Normalization.
> > - [3] QTSumm: Query-Focused Summarization over Tabular Data.
> >
> > Finally, we would like to sincerely thank the Reviewer once again for the valuable time and efforts. Please let us know if there is anything further we can do to address or clarify your concerns.

---

> > > ### Author Response · Authors · 2024-11-23
> > > **Regarding scalability of POS (cont)**
> > >
> > > Dear Reviewer `8Hmg `,
> > >
> > > Recognizing the importance of extending POS to **more dynamic or complex data environments where SQL alone may be insufficient**, we applied our method (POS) to the FeTaQA (Free-form Table Question Answering) dataset.
> > > FeTaQA involves free-form, natural language answers derived from tables, requiring nuanced reasoning across diverse contexts. In contrast, WikiTQ and TabFact primarily focus on data with less structural and contextual variability.
> > >
> > > On this dataset, we conducted comparisons between POS, End-to-End QA, and Few-Shot QA, similar to our evaluations on TabFact and WikiTQ. The results are reported as follows:
> > >
> > > Table: Evaluation on FeTaQA test set with gpt4o-mini.
> > > | Method             |    BLEU   |  ROUGE-1 |  ROUGE-L  |
> > > |--------------------|:---------:|:--------:|:---------:|
> > > | 0-shot QA   |   18.99   |   51.92  |   46.44   |
> > > | Few-shot QA |   19.18   |   53.32  |   46.86   |
> > > | Plan-of-SQLs (POS)      | **20.16** | **54.7** | **48.69** |
> > >
> > > We find that POS is better than End-to-End or Few-Shot QA in both aspects, accuracy and interpretability:
> > > - POS consistently achieves the best scores across all metrics on the FeTaQA test set, with a BLEU improvement of 0.98 pts and ROUGE-L gain of 1.83 pts over Few-shot QA. The reason for these improvements is that POS retrieves fine-grained and correct information from tables through its step-by-step SQLs, enabling better answer generation. Here we show an example of precise table information retrieval from POS.
> > >
> > > ```python
> > > Table Page Title: 1948 Oregon gubernatorial special election
> > > Table Section Title: Results
> > > ### Table:
> > > /*
> > > col : Party | Party | Candidate | Votes | %
> > > row 1 : - | Republican | Douglas McKay | 271,295 | 53.23
> > > row 2 : - | Democratic | Lew Wallace | 226,958 | 44.53
> > > row 3 : - | Independent | Wendell E. Barnett | 11,380 | 2.23
> > > row 4 : Total votes | Total votes | Total votes | 509,633 | 100
> > > row 5 : - | Republican hold | Republican hold | Republican hold | Republican hold
> > > */
> > > Question: By what percentage of votes did McKay win the election against Wallace's 44.53% vote?
> > > Plan:
> > > 1. Select rows where 'Candidate' is 'McKay' or 'Wallace'.
> > > 2. Select 'Candidate' and '%' columns.
> > > 3. Ask LLM to generate the final answer based on the current table.
> > > ```
> > > This information retrieval contrasts with End-to-End QA and Few-shot QA, which feed the entire large table into the LLM, making it harder for model to pinpoint relevant information and increasing the likelihood of errors.
> > >
> > > - POS provides visual explanations for the generated answers, which End-to-End QA and Few-shot QA lack. **We share an explanation from POS on FetaQA** [here](https://docs.google.com/presentation/d/1sefWsjA664LP9uhKAqDV2EtDpivK7O8h/edit?usp=sharing&ouid=106754253201604804412&rtpof=true&sd=true). As POS bases on a concise, final table to generate the final answer, this makes it easier for users to understand why the model generates a particular response. In contrast, End-to-End QA and Few-shot QA require users to refer back to the large, original table, making the reasoning process less understandable and harder to follow.
> > >
> > > As we near the end of the rebuttal window, we are eagerly looking forward to your acknowledgment.
> > > Thank you!

---

> ### Author Response · Authors · 2024-11-15
> **Regarding the novelty of POS**
>
> > Limited Novelty: While the paper addresses an important problem in
> > Table QA, the core ideas, such as breaking down queries into atomic
> > steps and using SQL transformations for interpretability, are established
> > in other domains. This reduces the method's originality, as it largely
> > adapts existing decomposition and programmatic interpretability
> > approaches rather than introducing fundamentally new techniques
> > specific to Table QA.
>
>
>
> Thank you for this insightful comment!
>
> Reviewer `8Hmg` said that:
> "the core ideas, such as breaking down queries into atomic steps and using SQL transformations for interpretability, are established in other domains."
>
> While it’s true that step-by-step planning is the main idea of almost all the LLM planning papers [a],
> to the best of our knowledge, **there is no such work in literature that do breaking down queries into atomic
> steps and using SQL transformations explicitly for interpretability**.
> If there are such papers, we would be grateful if the reviewer could point them out so that we can cite and give proper credit.
>
> We also acknowledge that our method leverages concepts from existing decomposition and programmatic interpretability approaches;
> however, we believe that POS introduces a novel application of these principles specifically tailored to Table QA.
>
> Besides, we would like to emphasize that our goal is not to introduce a new Table QA method that
> maximizes accuracy but rather to address critical interpretability gaps in current Table QA
> methods. Interpretability in this domain has been largely overlooked, and POS aims
> to fill that gap.
>
> To further support the interpretability frontier of POS, we conducted an additional evaluation where
> explanation methods (Text-to-SQL, DATER, CoTable, and POS) help us identify whether the Table QA model
> is correct (model prediction debugging[c,d,e,f]). Model prediction debugging is the mainstream and established task in XAI and has
> many real-world applications (e.g. healthcare[g]).
>
> Using both the TabFact and WikiTQ datasets, we evaluated the following question:
> Given the explanations, can users identify if the Table QA model is correct or wrong?
> We leverage LLM-as-a-Judge as we did in Forward Simulation and Preference Ranking for this setup.
>
> Here is our prompt to the LLM judges:
> ```
> The Table Question Answering (Table QA) model is working on a tabular dataset, answering questions based on a given table.
>
> You are given an HTML file containing a Question, Input Table, Prediction, and an Explanation clarifying the Prediction.
>
> Your task is to carefully analyze the explanation and determine whether the Prediction is correct or not.
>
> Explanation Method: [Text2SQL/DATER/CoTable/POS]
> Answer with ‘Correct’ or ‘Wrong’ only.
>
> You MUST ignore the order of the options and answer based on the correctness of the Prediction!
> ```
>
> We report the result of the evaluation below:
>
> Table: Model prediction debugging accuracy (&uarr;) on TabFact and WikiTQ with LLM judges using different Table QA explanations.
>
> |   Dataset   |   TabFact   |        |         |        | WikiTQ |        |
> |:-----------:|:-----------:|:------:|:-------:|:------:|:------:|:------:|
> |    Method   | Text-to-SQL | DATER  | CoTable |  POS   | DATER  |  POS   |
> | GPT-4o-mini |   55.37%    | 55.43% | 61.36%  | **76.74%** | 64.58% | **71.93%** |
> |    GPT-4o   |   55.97%    | 70.95% | 67.34%  | **72.85%** | 73.31% | **74.45%** |
> |    GPT-4    |   49.93%    | 57.56% | 60.38%  | **72.08%** | **73.5%**  | 72.38% |
>
> Our findings are:
> - POS significantly improves model prediction debugging accuracy compared to other Table QA baselines.
> - On the TabFact dataset, POS achieves a debugging accuracy of 76.74% with GPT-4o-mini, 72.85% with GPT-4o, and 72.08% with GPT-4, outperforming other methods by margins of up to 21%.
> - On WikiTQ, POS reaches the highest accuracy of 74.45 with GPT4-o.
>
> These results again confirm that POS offers a distinctive interpretability advantage and usefulness compared to existing Table QA models.
>
> References:
> - [a] Understanding the planning of LLM agents: A survey. 2024.
> - [b] Watch Your Steps: Observable and Modular Chains of Thought. 2024
> - [c] The effectiveness of feature attribution methods and its correlation with automatic evaluation scores, NeurIPS
> - [d] HIVE: Evaluating the Human Interpretability of Visual Explanations, ECCV.
> - [e] What I Cannot Predict, I Do Not Understand: A Human-Centered Evaluation Framework for Explainability Methods, NeurIPS.
> - [f] Debugging Tests for Model Explanations, NeurIPS.
> - [g] Post hoc Explanations may be Ineffective for Detecting Unknown Spurious Correlation, ICLR.
>
> We sincerely thank the Reviewer `8Hmg` once again for the insightful comments!

---

> ### Author Response · Authors · 2024-11-15
> **Regarding interpretability improvement**
>
> > Limited Effectiveness and Scalability: The performance improvements with
> > POS are incremental, indicating only a modest boost in interpretability
> > and accuracy over baseline methods.
>
> In this comment, Reviewer `8Hmg` mentioned that:
> "The performance improvements with POS are incremental, indicating only a modest boost in interpretability and accuracy over baseline methods."
> However, in the Strength 2nd, the Reviewer mentioned:
> "It includes a comprehensive evaluation, using both human and LLM judges to assess interpretability and predictive accuracy, showcasing POS’s advantages in clarity, coherence, and overall effectiveness compared to existing methods."
>
> We feel that there is mixed feedback on the interpretability of POS.
> As demonstrated in all 3 XAI tasks (Forward Simulation, Preference Ranking, and Model Prediction Debugging), POS consistently outperforms existing state-of-the-art Table QA methods in terms of interpretability and effectiveness.
>
> Please let us know if this address the Reviewer's concern about "modest boost in interpretability"!

---

> ### Comment · Reviewer_8Hmg · 2024-12-03
> **Response to the rebuttal**
>
> Thanks for your further clarification. It solves part of my concerns. However, I still do not think the Novelty of this paper reaches the standard of ICLR. As a result, I will keep my rating.

---

### Official Review · Reviewer_VQhj · 2024-10-30

**Soundness:** 3
**Presentation:** 2
**Contribution:** 3
**Rating:** 5
**Confidence:** 4

**Summary:**

The paper proposes a new in-context learning method for solving table-based QA, Plan-of-SQLs (PoS). The method first generates a sequence of atomic actions as the "plan" and translates the actions in the plan to simple SQL queries step by step. The final answer would be the result of the sequence of the translated SQL queries.

The author emphasizes the interpretability of the method by arguing all actions are taken through SQL queries and the output is the result of the SQL queries.

In terms of performance, the proposed method achieves comparable performance compared with baselines while is much better in interpretability.

**Strengths:**

The proposed method PoS learned many good merits from existing methods, including the plan generation, and step-by-step execution. It also proposed a novel point about the interpretability of Table QA. The explanation generation method is also novel where it highlights the related col/row/cells in the queries. This method is uniquely designed for Table QA.

**Weaknesses:**

1. Since this is an in-context learning method, the performance is greatly influenced by the choice of LLM backbones. The paper only uses gpt-3.5-turbo-16k-0613 which has already been deprecated by OpenAI. I recommend the authors choose more variety of models and use newer models. This would make the results more reliable and reproducible.

2. The XAI comparison is not very convincing. The highlight method is preferable for SQL-based methods such as PoS or Text-to-SQL. However, it is not explicit for other baselines without SQL, such as Dater and CoTable.

3. The authors argue that it is better for interpretability to merely depend on SQL to process the table and answer the question. However, the necessity of this constraint is still questionable for the table qa in practical scenarios. The tables are of various formats, and the answer usually cannot be produced by a SQL query. Such limitation may explain the lower performance of PoS compared with many other baselines.

4. The novelty of the method is limited. Since the step-by-step execution has been already proposed in CoTable, this work seems a simple extension of CoTable with SQL queries and plan generation. And the final performance is even reduced with the added components.

5. The paper's presentation requires improvement, as some tables extend beyond the page margins.

**Questions:**

1. Are there any cases where the answer cannot be generated by SQL queries?

2. What is the motivation to add highlights in the XAI experiments? Did you try to remove them in the experiments and see the results?

---

> ### Author Response · Authors · 2024-11-14
> **Thank You for Your Efforts and Thoughtful Reviews!**
>
> We sincerely appreciate Reviewer `VQhj`'s thoughtful reviews on our paper!
>
> Your comments, particularly regarding the **effect of highlights in XAI comparisons**, have provided us with valuable insights to improve our evaluation.
>
> Below, we show how your insights helped us improve evaluation and humbly answer your questions as follows:

---

> > ### Author Response · Authors · 2024-11-14
> > **Regarding the request to test POS on more LLM models**
> >
> > > Since this is an in-context learning method, the performance is greatly influenced by the choice of LLM backbones. The paper only uses gpt-3.5-turbo-16k-0613 which has already been deprecated by OpenAI. I recommend the authors choose more variety of models and use newer models. This would make the results more reliable and reproducible.
> >
> > Thank you for the suggestion to test POS on the newer LLM backbones!
> >
> > The reason we picked gpt-3.5-turbo-16k-0613 is because this LLM has
> > been widely used by a long line of works (CoTable, Binder, DATER,
> > TableSQLify) in Table QA. Using gpt-3.5-turbo-16k-0613 makes sure
> > we have a fair comparison between POS and baselines.
> > Given the deprecation of GPT3-5, we tested POS on GPT4o-mini and
> > report the numbers below:
> >
> > Table: Table QA accuracy of POS on TabFact and WikiTQ with GPT3-5 and GPT4o-mini
> >
> > |            |    TabFact    |        |     WikiTQ    |        |
> > |:----------:|:-------------:|:------:|:-------------:|:------:|
> > |   Method   | End-to-end QA |   POS  | End-to-end QA |   POS  |
> > |   GPT3-5   |     70.45%    | 78.31% |     51.84%    | 54.80% |
> > | GPT4o-mini |     71.17%    | 83.45% |     49.24%    | 54.95% |
> >
> > We plan to add GPT4o and GPT4-turbo in the next version of the paper.

---

> ### Author Response · Authors · 2024-11-14
> **Regarding technical novelty of POS**
>
> > The novelty of the method is limited. Since the step-by-step execution has been already proposed in CoTable, this work seems a simple extension of CoTable with SQL queries and plan generation. And the final performance is even reduced with the added components.
>
> Thank you for your comment!
> While POS shares similarities with CoTable in its step-by-step approach,
> we believe our method introduces key innovations beyond CoTable. POS systematically decomposes
> complex queries into atomic, interpretable sub-steps and SQL-compatible transformations,
> providing transparency that CoTable and other existing methods lack. Unlike traditional
> program-based approaches, which often struggle with unexplainable selections or function
> arguments (see Fig. 1), POS allows each transformation to be fully traceable, making
> the entire reasoning process interpretable.
>
> We would like to emphasize that **our goal is** NOT to improve Table QA accuracy but rather **to improve the interpretability of Table QA responses** so that users and other LLM agents could make better decisions.
> This is because understanding how a model arrives at a decision in a information-dense format as a table poses a challenge to the cognitive load of users to verify the predicted answer and identify errors in the reasoning chain.
>
> To further support the interpretability frontier of POS, we conducted an additional evaluation where
> explanation methods (Text-to-SQL, DATER, CoTable, and POS) help us **identify whether the predicted answer by LLM
> is correct** (model prediction debugging[c,d,e,f]). Model prediction debugging is the mainstream and established task in XAI and has
> many real-world applications (e.g. healthcare[g]).
>
> Using both the TabFact and WikiTQ datasets, we evaluated the following question:
> Given the explanations, can users identify if the predicted LLM answer is correct or wrong?
> We leverage LLM-as-a-Judge as we did in Forward Simulation and Preference Ranking for this setup.
>
> Here is our prompt to the LLM judges:
> ```
> The Table Question Answering (Table QA) model is working on a tabular dataset, answering questions based on a given table.
>
> You are given an HTML file containing a Question, Input Table, Prediction, and an Explanation clarifying the Prediction.
>
> Your task is to carefully analyze the explanation and determine whether the Prediction is correct or not.
>
> Explanation Method: [Text2SQL/DATER/CoTable/POS]
> Answer with ‘Correct’ or ‘Wrong’ only.
>
> You MUST ignore the order of the options and answer based on the correctness of the Prediction!
> ```
>
> We report the result of the evaluation below:
>
> Table: Model prediction debugging accuracy (&uarr;) on TabFact and WikiTQ with LLM judges using different Table QA explanations.
>
> |   Dataset   |   TabFact   |        |         |        | WikiTQ |        |
> |:-----------:|:-----------:|:------:|:-------:|:------:|:------:|:------:|
> |    Method   | Text-to-SQL | DATER  | CoTable |  POS   | DATER  |  POS   |
> | GPT-4o-mini |   55.37%    | 55.43% | 61.36%  | **76.74%** | 64.58% | **71.93%** |
> |    GPT-4o   |   55.97%    | 70.95% | 67.34%  | **72.85%** | 73.31% | **74.45%** |
> |    GPT-4    |   49.93%    | 57.56% | 60.38%  | **72.08%** | **73.5%**  | 72.38% |
>
> Our findings are:
> - POS significantly improves model prediction debugging accuracy compared to other Table QA baselines.
> - On the TabFact dataset, POS achieves a debugging accuracy of `76.74%` with GPT-4o-mini, `72.85%` with GPT-4o, and `72.08%` with GPT-4, outperforming other methods by margins of up to `21%`.
> - On WikiTQ, POS reaches the highest accuracy of `74.45%` with GPT4-o.
>
> These results again confirm that POS offers a distinctive interpretability advantage and usefulness compared to existing Table QA models.
>
> References:
> - [c] The effectiveness of feature attribution methods and its correlation with automatic evaluation scores, NeurIPS
>
> - [d] HIVE: Evaluating the Human Interpretability of Visual Explanations, ECCV.
>
> - [e] What I Cannot Predict, I Do Not Understand: A Human-Centered Evaluation Framework for Explainability Methods, NeurIPS.
>
> - [f] Debugging Tests for Model Explanations, NeurIPS.
>
> - [g] Post hoc Explanations may be Ineffective for Detecting Unknown Spurious Correlation, ICLR.

---

> ### Author Response · Authors · 2024-11-14
> **Regarding the motivation of adding highlights in explanation**
>
> > What is the motivation to add highlights in the XAI experiments?
>
> Thank you for this great question!
>
> Highlights (or attribution maps) are widely recognized as a key method for explaining
> AI decisions to humans across all domains, including images[a], text[b], and time series[c].
> In these domains, highlights effectively draw users’ attention to the most relevant parts of
> the data, helping them understand how the model arrived at its decisions.
>
> - [a] What i cannot predict, i do not understand: A human-centered evaluation framework for explainability methods, NeurIPS
> - [b] Evaluating Explainable AI: Which Algorithmic Explanations Help Users Predict Model Behavior?, ACL
> - [c] Explainable AI for Time Series Classification, IEEE Access
>
> For Table QA, there has been limited work visualizing model explanations in an interpretable way.
> To address this gap, we adapted the highlight method from other domains for tabular data
> to help users understand how the model utilizes specific parts (features) of the input table
> to arrive at the final answer. By highlighting relevant rows, columns, or cells in the table,
> we aim to provide a clear visual explanation of the model's reasoning process.

---

> ### Author Response · Authors · 2024-11-14
> **Regarding the effects in highlights in XAI comparison**
>
> > The XAI comparison is not very convincing. The highlight method is preferable for SQL-based methods such as PoS or Text-to-SQL. However, it is not explicit for other baselines without SQL, such as Dater and CoTable.
> Did you try to remove them in the experiments and see the results?
>
> We find this comment very thought-provoking, and it indeed aided us in improving our XAI evaluation.
>
> We understand that the highlight-based interpretability
> method may naturally align better with SQL-based approaches like POS and Text-to-SQL.
> For non-SQL methods, such as Dater and CoTable, we agree that highlight-based interpretability
> may be less advantageous compared to SQL-based methods.
>
> Following Reviewer `VQhj`'s suggestion, we removed highlights to assess which explanations
> perform best without the potential advantage that highlights may give to methods like
> Text-to-SQL or POS.
>
> We tested explanations with no highlights on three XAI tasks: Preference Ranking, Forward Simulation, and [Model Debugging](https://openreview.net/forum?id=mDV36U4d6u&noteId=3PACyF2oBr)
> (an additional, more standard XAI evaluation).
> We repeat the LLM-judge setups as described in our paper for GPT4o-mini
> and report the data below:
>
> Table: Evaluating Table QA explanations with no highlights (attribution maps).
>
> |     Task    | Preference Ranking (↓) | Forward Simulation (↑) | Model Debugging (↑) |
> |:-----------:|:----------------------:|:----------------------:|:-------------------:|
> | Text-to-SQL |          3.97          |         65.67%         |        55.37%       |
> |    DATER    |          2.64          |         73.86%         |        56.32%        |
> |   CoTable   |          1.76          |         75.94%         |        64.18%        |
> |  POS (Ours) |          1.62          |         81.81%         |        75.10%       |
>
> Our findings are:
> - Without highlights, **POS still performs best** across all three XAI tasks—Preference Ranking, Forward Simulation, and Model Debugging.
> - The removal of highlights does not affect the accuracy/ratings for the explanation methods, suggesting that highlight-based explanations may not be important for LLMs perhaps because it is unknown which formats of highlights (image/xml tags/mark-up syntax) should be effective for language models. This insight could guide future developments in XAI methods for Table QA.
>
> References:
>
> - [a] The effectiveness of feature attribution methods and its correlation with automatic evaluation scores, NeurIPS
> - [b] What i cannot predict, i do not understand: A human-centered evaluation framework for explainability methods, NeurIPS

---

> ### Author Response · Authors · 2024-11-14
> **Regarding the necessity of SQL-only constraints in the tasks**
>
> > The authors argue that it is better for interpretability to merely depend on SQL to process the table and answer the question. However, the necessity of this constraint is still questionable for the table qa in practical scenarios. The tables are of various formats, and the answer usually cannot be produced by a SQL query. Such limitation may explain the lower performance of PoS compared with many other baselines.
> Are there any cases where the answer cannot be generated by SQL queries?
>
> Thank you for this great comment!
>
> Yes, there are cases where the answer cannot be generated by SQL queries alone, particularly those involving long-form generation—such as text summarization tasks in datasets like QTSumm [a]—often relying on model “creativity” to produce answers. For such datasets, relying solely on SQL is not optimal. We argue that combining the strengths of SQL for structured data handling with the creative reasoning capabilities of LLMs offer a more powerful approach.
>
> We recognize that tables vary significantly in format, which can sometimes make SQL-based processing challenging [b]. However, by preprocessing samples into a SQL-compatible format (i.e., making them "SQLifiable"), we can achieve a significantly better interpretable decision-making process. For instance, the NormTab paper [c] demonstrates effective data cleaning techniques to prepare tables for SQL processing, showing that, after cleaning, applying naive methods like Text-to-SQL can boost accuracy on datasets like TabFact and WikiTQ by 5-10 points. This shows that POS can not only provide great interpretability but also accurate answers.
>
> We also want to note that our primary focus in this work is on interpretability, which is why we emphasize SQL-based methods. For long-form generation, hybrid explanation methods (e.g. SQL + natural-language explanations) would be more promising and we leave this exploration for future work. We hope POS lays the groundwork for interpretability research in Table QA.
> - [a] QTSumm: Query-Focused Summarization over Tabular Data, EMNLP
> - [b] On the potential of lexico-logical alignments for semantic parsing to sql queries, EMNLP
> - [c] NormTab: Improving Symbolic Reasoning in LLMs Through Tabular Data Normalization, EMNLP

---

> > ### Author Response · Authors · 2024-11-23
> > **Regarding the necessity of SQL-only constraints in the tasks (cont)**
> >
> > Dear Reviewer `VQhj `,
> >
> > To further address your concerns about the scalability of POS in practical QA scenarios, we applied our method (POS) to the FeTaQA (Free-form Table Question Answering) dataset.
> > FeTaQA involves free-form, natural language answers derived from tables, requiring nuanced reasoning across diverse contexts. In contrast, WikiTQ and TabFact primarily focus on data with less structural and contextual variability.
> >
> > On this dataset, we conducted comparisons between POS, End-to-End QA, and Few-Shot QA, similar to our evaluations on TabFact and WikiTQ. The results are reported as follows:
> >
> > Table: Evaluation on FeTaQA test set with gpt4o-mini.
> > | Method             |    BLEU   |  ROUGE-1 |  ROUGE-L  |
> > |--------------------|:---------:|:--------:|:---------:|
> > | 0-shot QA   |   18.99   |   51.92  |   46.44   |
> > | Few-shot QA |   19.18   |   53.32  |   46.86   |
> > | Plan-of-SQLs (POS)      | **20.16** | **54.7** | **48.69** |
> >
> > We find that POS is better than End-to-End or Few-Shot QA in both aspects, accuracy and interpretability:
> > - POS consistently achieves the best scores across all metrics on the FeTaQA test set, with a BLEU improvement of 0.98 pts and ROUGE-L gain of 1.83 pts over Few-shot QA. The reason for these improvements is that POS retrieves fine-grained and correct information from tables through its step-by-step SQLs, enabling better answer generation. Here we show an example of precise table information retrieval from POS.
> >
> > ```python
> > Table Page Title: 1948 Oregon gubernatorial special election
> > Table Section Title: Results
> > ### Table:
> > /*
> > col : Party | Party | Candidate | Votes | %
> > row 1 : - | Republican | Douglas McKay | 271,295 | 53.23
> > row 2 : - | Democratic | Lew Wallace | 226,958 | 44.53
> > row 3 : - | Independent | Wendell E. Barnett | 11,380 | 2.23
> > row 4 : Total votes | Total votes | Total votes | 509,633 | 100
> > row 5 : - | Republican hold | Republican hold | Republican hold | Republican hold
> > */
> > Question: By what percentage of votes did McKay win the election against Wallace's 44.53% vote?
> > Plan:
> > 1. Select rows where 'Candidate' is 'McKay' or 'Wallace'.
> > 2. Select 'Candidate' and '%' columns.
> > 3. Ask LLM to generate the final answer based on the current table.
> > ```
> > This information retrieval contrasts with End-to-End QA and Few-shot QA, which feed the entire large table into the LLM, making it harder for model to pinpoint relevant information and increasing the likelihood of errors.
> >
> > - POS provides visual explanations for the generated answers, which End-to-End QA and Few-shot QA lack. **We share an explanation from POS on FetaQA** [here](https://docs.google.com/presentation/d/1sefWsjA664LP9uhKAqDV2EtDpivK7O8h/edit?usp=sharing&ouid=106754253201604804412&rtpof=true&sd=true). As POS bases on a concise, final table to generate the final answer, this makes it easier for users to understand why the model generates a particular response. In contrast, End-to-End QA and Few-shot QA require users to refer back to the large, original table, making the reasoning process less understandable and harder to follow.
> >
> > As we near the end of the rebuttal window, we are eagerly looking forward to your acknowledgment.
> > Thank you!

---

> > > ### Comment · Reviewer_VQhj · 2024-11-29
> > >
> > > Thank you for the comprehensive response to my review. I will raise my score to 5 while I am still not very convinced by the motivation about the interpretability of table understanding.

---

> > > > ### Author Response · Authors · 2024-11-30
> > > > **Thank you for your review and feedback!**
> > > >
> > > > Thank you Reviewer `VQhj`,
> > > >
> > > > We really appreciate your time reading our comprehensive rebuttal!

---

> ### Author Response · Authors · 2024-11-14
> **Regarding paper format**
>
> > The paper's presentation requires improvement, as some tables extend beyond the page margins
>
> Thank you for bringing this to our attention! We will ensure that all tables are properly formatted within the page margins in the next revision.
>
> If there are any further concerns, we would be more than happy to address them and continue the discussion. We are excited to engage in any way!

---

### Official Review · Reviewer_mQDc · 2024-10-30

**Soundness:** 3
**Presentation:** 3
**Contribution:** 2
**Rating:** 6
**Confidence:** 4

**Summary:**

This paper presents Plan-of-SQLs (POS), an interpretable Table Question Answering (Table QA) method that enhances model transparency by decomposing complex queries into atomic natural-language sub-queries. Each sub-query is sequentially converted into SQL commands, allowing the input table to be transformed step-by-step until the final answer is obtained. This decomposition approach ensures that every transformation is clear and traceable, providing a fully interpretable reasoning process. POS achieves competitive performance on standard Table QA benchmarks (TabFact and WikiTQ) while significantly improving clarity, coherence, and helpfulness compared to existing models such as CoTable and DATER.

**Strengths:**

The paper is well-structured and written, with a logical flow that makes the methodology and experiments easy to follow. The proposed POS advances interpretability in Table QA by decomposing complex queries into atomic, natural-language sub-queries sequentially converted into SQL commands. The authors support their interpretability claims with comprehensive experiments, including both human and LLM evaluations, showcasing the clarity, coherence, and helpfulness of POS explanations.

**Weaknesses:**

While POS enhances interpretability, its innovation appears incremental, with its main contribution being the decomposition of queries into SQL-translatable atomic steps. Furthermore, POS shows limited accuracy improvements over existing models, which could diminish its attractiveness in applications where interpretability is less prioritized. Another concern is POS’s reliance on sequential processing via the NL Atomic Planner, where each plan requires the previous step's results, including intermediate table states. This dependence on continuous LLM calls may lead to inefficiencies, particularly for complex or large tables, as each step adds computational overhead. Detailed efficiency comparisons with non-sequential methods or potential optimizations would strengthen the paper and address this limitation.

**Questions:**

1.	Given POS’s sequential nature and reliance on intermediate tables as inputs for each step, what are the efficiency implications?
2.	In Appendix C, the ablation study discusses changes in interpretability after removing different modules, but no quantitative metrics are provided to measure these changes. Could the authors include specific interpretability metrics to quantify the impact of each module?
3.	There appear to be issues with Figure 6. The function parameters in Step 1 are incorrect, and there is an unintended split in the image for Step 3.

---

> ### Author Response · Authors · 2024-11-14
> **Thank You for Your Positive Feedback and Insightful Suggestions!**
>
> We sincerely thank the reviewer `mQDc` for the feedback on our work and for recognizing POS’s contributions to interpretability in Table QA.
>
> We particularly appreciate the suggestion regarding efficiency analysis, which has helped us better showcase the efficiency advantages of our method.
>
> Below, we detail how your insights guided us in conducting this analysis and addressing your questions!

---

> > ### Author Response · Authors · 2024-11-14
> > **Regarding Figure 6 issues**
> >
> > > 3. There appear to be issues with Figure 6. The function parameters in Step 1 are incorrect, and there is an unintended split in the image for Step 3.
> >
> > Thank you for this detailed comment!
> >
> > In Fig.6, we actually visualize exactly the row indices given by the Chain-of-Table method (i.e. row 1, row 2, row 3, row 4, row 8). In Chain-of-Table, it ignores the header row (column names), and start with index 0 (here refers to game 1). Please note that in human studies, we also explicitly told users that CoTable row indices start from 0 and we do not count the header row.
> >
> > However, we agree with the Reviewer that we can adjust these indices to make the visualization less confusing and aligned with other methods (e.g. DATER in Fig.5). Regarding the unintended split, this was an error in converting the visualization to PDF for insertion into Overleaf. We will address both issues in the next revision.
> >
> > Thank you once again for the positive and constructive feedback!

---

> ### Author Response · Authors · 2024-11-14
> **Regarding POS efficiency analysis**
>
> > 1. Given POS’s sequential nature and reliance on intermediate tables as inputs for each step, what are the efficiency implications?
>
> Thank you for this valuable question!
>
> To analyze the efficiency of POS, we followed the analysis used in the
> Chain-of-Table paper (Sec 4.5), using the total number of LLM queries as the measure of efficiency.
>
> When comparing POS to other baselines, including Binder (Cheng et al., 2022),
> Dater (Ye et al., 2023), and CoTable (Wang et al., 2024), we observe a distinct advantage:
> POS does not rely on self-consistency (generating multiple candidate answers and selecting the most frequent or consistent one to improve accuracy)
> during either the planning or SQL generation steps,
> whereas all three baseline methods use self-consistency to boost accuracy.
> This helps POS significantly reduce the number of LLM queries needed per sample (to only `4`),
> making POS the most efficient among these baselines. See the Table below.
>
> Table: Efficiency analysis on WikiTQ
>
> | Method                      | Self-consistency | LLM queries | Breakdown LLM queries                                                | Database queries |
> |-----------------------------|------------------|-------------|----------------------------------------------------------------------|------------------|
> | Binder (Cheng et al., 2022) | Yes              | 50          | Generate Neural-SQL: 50                                              | 50               |
> | Dater (Ye et al., 2023)     | Yes              | 100         | Decompose Table: 40; Generate Cloze: 20; Generate SQL: 20; Query: 20 | 20               |
> | CoTable (Wang et al., 2024) | Yes              | ≤25         | DynamicPlan: ≤5; GenerateArgs: ≤19; Query: 1                         | 5                |
> | Plan-of-SQLs (Ours)         | No               | 4           | Planning: 2 Generate SQL: 2                                          | 2                |
>
> Additionally, we propose to compare the efficiency of these methods based on the number of
> table transformations, as a higher count leads to increased workload on the local database
> when handling Table QA tasks (i.e., higher number of queries to the table database).
>
> In this regard, POS also demonstrates a clear advantage over the baselines.
> In particular, while methods like Chain-of-Table rely on a predefined set of `5`
> operations (e.g., add column, select column/row, group column, sort column) to maintain
> consistency, POS requires only `2` database queries on average, making it the most efficient
> approach in terms of database queries. Similarly, Binder and Dater require
> significantly greater numbers of operations, with Binder doing table database queries `50` times
> and Dater sending slot-filling queries `20` times.

---

> ### Author Response · Authors · 2024-11-14
> **Regarding ablation study for interpretability**
>
> > 2. In Appendix C, the ablation study discusses changes in interpretability after removing different modules, but no quantitative metrics are provided to measure these changes. Could the authors include specific interpretability metrics to quantify the impact of each module?
>
> Thank you for highlighting this point!
>
> We would like to clarify that the goal of the ablation study in Appendix C is to test how each component affects POS QA **accuracy**, not interpretability.
>
> However, to address interpretability quantitatively, we can use the fallback rate as a proxy
> metric. In our approach, fallback occurs when an error in the pipeline occurs (e.g. due to an unexecutable SQL action) and we send the entire data to LLMs to get an answer. This approach however lacks interpretability (as shown in Fig. 1). Thus, a lower fallback rate indicates that POS can solve more queries using SQL, providing a fully interpretable decision-making process.
>
> As shown in Table 4, modules like NL Planning and Text-to-SQL significantly contribute to
> interpretability, with their removal increasing the fallback rate to 40-50%.
> Removing NL Planning results in SQL-only steps (in lieu of natural-language steps Fig. 7) will
> diminish interpretability, as shown by lower performance of Text-to-SQL compared to POS in the Forward Simulation experiment (Table 1).
>
> Similarly, without Text-to-SQL, LLM performs table transformations directly in natural language without using SQLs, which is less interpretable due to the inherent LLM black-box reasoning.
>
> When the Atomicity is ablated out, the fallback rate does not change much. To justify the change in interpretability, we visualized the plans generated by NL Planner in Fig.8 and Fig.9 and conducted a qualitative analysis (on 200 samples), which hinted that explanations were indeed less interpretable in this setup due to complex steps (see Fig.8 and 9’s captions).

---

> ### Author Response · Authors · 2024-11-15
> **Regarding ablation study for interpretability (continued)**
>
> As suggested by Reviewer `VQhj`, we conducted another ablation study on interpretability .
>
> In our paper, explanations consist of attribution maps (highlights) combined with natural-language steps to (for POS) or functions (for DATER and CoTable) or SQL command (for Text-to-SQL). To assess the impact of attribution maps, we removed highlights in explanations and give to XAI judges.
>
> We tested explanations with no highlights on three XAI tasks: Preference Ranking, Forward Simulation, and [Model Debugging](https://openreview.net/forum?id=mDV36U4d6u&noteId=6fDz8ayF1w)
> (an additional, more standard XAI evaluation).
> We repeat the LLM-judge setups as described in our paper for GPT4o-mini
> and report the data below:
>
> Table: Evaluating Table QA explanations with no highlights (attribution maps).
>
> |     Task    | Preference Ranking (↓) | Forward Simulation (↑) | Model Debugging (↑) |
> |:-----------:|:----------------------:|:----------------------:|:-------------------:|
> | Text-to-SQL |          3.97          |         65.67%         |        55.37%       |
> |    DATER    |          2.64          |         73.86%         |        56.32%        |
> |   CoTable   |          1.76          |         75.94%         |        64.18%        |
> |  POS (Ours) |          1.62          |         81.81%         |        75.10%       |
>
> Our findings are:
> - Without highlights, `POS still performs best` across all three XAI tasks—Preference Ranking, Forward Simulation, and Model Debugging.
> - The removal of highlights does not affect the accuracy/ratings for the explanation methods, suggesting that highlight-based explanations may not be important for LLMs perhaps because it is unknown which formats of highlights (image/xml tags/mark-up syntax) should be effective for language models. This insight could guide future developments in XAI methods for Table QA.
>
> References:
>
> - [a] The effectiveness of feature attribution methods and its correlation with automatic evaluation scores, NeurIPS
> - [b] What i cannot predict, i do not understand: A human-centered evaluation framework for explainability methods, NeurIPS

---

### Official Review · Reviewer_j2Vr · 2024-11-02

**Soundness:** 3
**Presentation:** 3
**Contribution:** 2
**Rating:** 5
**Confidence:** 5

**Summary:**

This paper introduces Plan-of-SQLs (POS), an approach for enhancing interpretability in Table Question Answering. POS decomposes complex queries into atomic sub-queries, each translated into SQL commands executed sequentially. This design enables users to trace the decision-making process step-by-step. The authors claim that POS not only outperforms comparable Table QA methods in interpretability but also maintains competitive accuracy, with promising results on benchmarks such as TabFact and WikiTQ.

**Strengths:**

- POS is well-designed for interpretability by breaking down complex queries into simple, understandable natural language planning. This sequential process makes it easier for users to follow and verify each stage of the answer generation.
- The authors employ both subjective and objective evaluations, involving human and LLM judges, to assess POS’s interpretability, coherence, and helpfulness, providing strong evidence for POS's improvement on interpretability.
- POS performs reasonably well on key benchmarks, achieving high interpretability without compromising accuracy.

**Weaknesses:**

- The authors report that 9.8% of samples on TabFact and 27.8% on WikiTQ could not be fully processed using POS and thus defaulted to end-to-end QA methods. Even on relatively simple Table QA datasets like TabFact and WikiTQ, these rates are notably high, raising concerns about POS’s scalability to more complex datasets, such as TabMWP. If the unprocessable rate rises significantly with more complex datasets, it raises the concern that POS may only improve interpretability with a sacrifice of precision provided in pure program-based methods.  The authors do not provide enough analysis regarding this matter, such as a comparison of unprocessable rates between POS and other program-based models and the unprocessable rates on more complex table-QA datasets.

- The technical contribution of POS is limited, since the whole framework primarily involves prompt engineering. Essentially, compared to traditional program-based methods, POS simply adds an additional layer of prompts to generate natural language “plans” that function as pseudo-comments for SQL statements. This does not introduce substantial technical contribution beyond traditional program-based methods.

- The NL Atomic Planner in POS depends on in-context planning examples that are specifically tailored to each dataset. This raises questions about its adaptability to a variety of reasoning types and Table QA problems. It remains unclear whether the current prompting schema can generalize effectively across different Table QA tasks. Further experimentation across a broader range of datasets (ideally 5–6) would be needed to demonstrate POS’s generalizability and robustness.

**Questions:**

Please refer to the weaknesses part.

---

> ### Author Response · Authors · 2024-11-14
> **Thank You for Your Thoughtful Review and Feedback!**
>
> We sincerely appreciate Reviewer `j2Vr`'s thoughtful feedback of our paper! Your comments, particularly on the `unprocessable rate` and the `generalizability` of in-context planning examples, provided valuable insights. We detail how your insights help us answer to your concerns below!

---

> > ### Author Response · Authors · 2024-11-14
> > **Regarding unprocessable rate of POS**
> >
> > > 1. The authors report that 9.8% of samples on TabFact and 27.8% on WikiTQ could not be fully processed using POS and thus defaulted to end-to-end QA methods. Even on relatively simple Table QA datasets like TabFact and WikiTQ, these rates are notably high, raising concerns about POS’s scalability to more complex datasets, such as TabMWP. If the unprocessable rate rises significantly with more complex datasets, it raises the concern that POS may only improve interpretability with a sacrifice of precision provided in pure program-based methods. The authors do not provide enough analysis regarding this matter, such as a comparison of unprocessable rates between POS and other program-based models and the unprocessable rates on more complex table-QA datasets.
> >
> >
> > Thank you for the great feedback!
> > As Reviewer `j2Vr` noted, the high rate of unprocessable samples in POS impacts scalability, particularly with complex datasets.
> >
> > Encouraged by this feedback, we delved into all cases where samples were unprocessable by POS (9.8% on TabFact and 27.8% on WikiTQ) and identified that **one-time planning** (generating a complete plan of steps at once in the beginning) is the bottleneck.
> > Here is an example where one-time planning made the sample unprocessable:
> >
> > **Query**: True or False? No games were played in **december**
> >
> > Table caption: 2011 final games in mlb
> >
> > | game_id |   game_date  | score | attendance | team_name |
> > |:-------:|:------------:|:-----:|:----------:|:---------:|
> > |    1    |  5 nov 2011  |   3   |    2000    |  falcons  |
> > |    2    |  12 nov 2011 |   4   |    1500    |   hawks   |
> > |    3    |  21 oct 2111 |   2   |    1800    |   eagles  |
> > |    4    | 30 sept 2011 |   1   |    2200    |   bears   |
> > |    5    |  20 nov 2011 |   3   |    2100    |  wildcats |
> >
> > Plan:
> >
> > 1. Extract the month from the **date** column then add a new column.
> > 2. Select rows where **month = 12**.
> > 3. Use a **CASE** statement to return **TRUE** if the number of rows is **0**, otherwise **FALSE**.
> >
> > In this example, the NL Planner generated Step 2 without seeing the output table of Step 1, which added a column with month names as text (e.g., "dec") instead of numeric values (e.g., "12").
> > Additionally, there is no guarantee that the new column from Step 1 is named "month," potentially causing Step 2 to reference a non-existent column.
> > We found `198` such cases in TabFact and `1207` in WikiTQ, where the one-time planning approach makes Table QA queries unprocessable.
> >
> > In one-time planning (generating a complete plan of steps at once in the beginning), the NL Planner assumes that previous transformations have succeeded and produced the expected output. If any step fails (e.g., due to missing columns or data formatting issues), the process halts, leading to an unprocessable sample.
> >
> > We solved this issue by `step-by-step planning`—prompting the NL Planner to generate one step at a time (instead of generating a complete plan at once), with the input of NL Planner being the previous steps and the current intermediate table.
> > As shown in the tables below, step-by-step planning led to substantial improvements in processing accuracy and unprocessable rates for both TabFact and WikiTQ.
> >
> > Table: Table QA accuracy (&uarr;) on TabFact and WikiTQ datasets using step-by-step planning
> >
> > | Method                    | TabFact         | WikiTQ          |
> > |---------------------------|-----------------|-----------------|
> > | End-to-end QA             | 71.17%          | 49.24%          |
> > | POS one-time planning     | 77.22%          | 48.90%          |
> > | POS step-by-step planning | 83.45% (+6.23%) | 54.95% (+6.05%) |
> >
> >
> > Table: Unprocessable rates (&darr;) on TabFact and WikiTQ datasets using step-by-step planning.
> >
> > | Method                    | TabFact       | WikiTQ         |
> > |---------------------------|---------------|----------------|
> > | POS one-time planning     | 10.56%        | 22.63%         |
> > | POS step-by-step planning | 3.16% (-7.4%) | 13.58 (-9.05%) |
> >
> > Using GPT4o-mini as the LLM backbone, we found that POS is able to process the vast majority
> > of examples using SQLs only, reaching up to about `97%` on TabFact and `86%` on WikiTQ!
> > It is important to note that WikiTQ contains considerable data noise [a], thus achieving this processable rate using SQLs alone is non-trivial.
> >
> > Again, we appreciate the Reviewer’s insights, which guided us toward a more robust and scalable solution for handling the problem.
> >
> > Regarding the scalability of POS to more complex datasets like TabMWP [b]. After looking at the dataset, we acknowledge that TabMWP indeed presents additional challenges due to its complexity and the diversity of reasoning requirements.
> > We will ensure that our paper includes a discussion on the potential and limitations of scaling POS to handle more complex problems.
> >
> > - [a] On the potential of lexico-logical alignments for semantic parsing to sql queries
> >
> > - [b] Dynamic Prompt Learning via Policy Gradient for Semi-structured Mathematical Reasoning

---

> ### Author Response · Authors · 2024-11-14
> **Regarding POS technical contribution**
>
> > 2. The technical contribution of POS is limited, since the whole framework primarily involves prompt engineering. Essentially, compared to traditional program-based methods, POS simply adds an additional layer of prompts to generate natural language “plans” that function as pseudo-comments for SQL statements. This does not introduce substantial technical contribution beyond traditional program-based methods.
>
> Thank you for your comment! While POS does indeed utilize prompt engineering, its design
> extends beyond simple prompt layering. The key innovation lies in its systematic decomposition
> of complex queries into atomic, **interpretable** sub-steps and table transformations—a feature not found in
> existing approaches. Unlike traditional program-based methods, which often struggle with unexplainable selections or function arguments (Fig. 1),
> POS offers a fully transparent, step-by-step transformation process.
>
> We would like to emphasize that our goal is not to introduce a new Table QA method that
> maximizes accuracy but rather to address critical interpretability gaps in current Table QA
> methods. Interpretability in this domain has been largely overlooked, and POS aims
> to fill that gap.
>
> To further support the interpretability frontier of POS, we conducted an additional evaluation where
> explanation methods (Text-to-SQL, DATER, CoTable, and POS) help us identify whether the Table QA model
> is correct (model prediction debugging[c,d,e,f]). Model prediction debugging is the mainstream and established task in XAI and has
> many real-world applications (e.g. healthcare[g]).
>
> Using both the TabFact and WikiTQ datasets, we evaluated the following question:
> Given the explanations, can users identify if the Table QA model is correct or wrong?
> We leverage LLM-as-a-Judge as we did in Forward Simulation and Preference Ranking for this setup.
>
> Here is our prompt to the LLM judges:
> ```
> The Table Question Answering (Table QA) model is working on a tabular dataset, answering questions based on a given table.
>
> You are given an HTML file containing a Question, Input Table, Prediction, and an Explanation clarifying the Prediction.
>
> Your task is to carefully analyze the explanation and determine whether the Prediction is correct or not.
>
> Explanation Method: [Text2SQL/DATER/CoTable/POS]
> Answer with ‘Correct’ or ‘Wrong’ only.
>
> You MUST ignore the order of the options and answer based on the correctness of the Prediction!
> ```
>
> We report the result of the evaluation below:
>
> Table: Model prediction debugging accuracy (&uarr;) on TabFact and WikiTQ with LLM judges using different Table QA explanations.
>
> |   Dataset   |   TabFact   |        |         |        | WikiTQ |        |
> |:-----------:|:-----------:|:------:|:-------:|:------:|:------:|:------:|
> |    Method   | Text-to-SQL | DATER  | CoTable |  POS   | DATER  |  POS   |
> | GPT-4o-mini |   55.37%    | 55.43% | 61.36%  | **76.74%** | 64.58% | **71.93%** |
> |    GPT-4o   |   55.97%    | 70.95% | 67.34%  | **72.85%** | 73.31% | **74.45%** |
> |    GPT-4    |   49.93%    | 57.56% | 60.38%  | **72.08%** | **73.5%**  | 72.38% |
>
> Our findings are:
> - POS significantly improves model prediction debugging accuracy compared to other Table QA baselines.
> - On the TabFact dataset, POS achieves a debugging accuracy of 76.74% with GPT-4o-mini, 72.85% with GPT-4o, and 72.08% with GPT-4, outperforming other methods by margins of up to 21%.
> - On WikiTQ, POS reaches the highest accuracy of 74.45 with GPT4-o.
>
> These results again confirm that POS offers a distinctive interpretability advantage and usefulness compared to existing Table QA models.
>
> References:
> - [c] The effectiveness of feature attribution methods and its correlation with automatic evaluation scores, NeurIPS
>
> - [d] HIVE: Evaluating the Human Interpretability of Visual Explanations, ECCV.
>
> - [e] What I Cannot Predict, I Do Not Understand: A Human-Centered Evaluation Framework for Explainability Methods, NeurIPS.
>
> - [f] Debugging Tests for Model Explanations, NeurIPS.
>
> - [g] Post hoc Explanations may be Ineffective for Detecting Unknown Spurious Correlation, ICLR.

---

> ### Author Response · Authors · 2024-11-14
> **Regarding the generalizability of in-context examples**
>
> > 3. The NL Atomic Planner in POS depends on in-context planning examples that are specifically tailored to each dataset. This raises questions about its adaptability to a variety of reasoning types and Table QA problems. It remains unclear whether the current prompting schema can generalize effectively across different Table QA tasks.
>
> Thank you for the insightful comment!
>
> Our in-context planning examples were indeed crafted specifically for each dataset.
> To explore the generalizability of our prompting schema, we modified the examples
> to remove any dataset-specific details. These revised, `general` in-context examples are
> designed to handle typical Table QA tasks, where the input to the NL Planner
> is simply the Query/Table and the output is the Plan.
>
> An example of the new generalized in-context prompt is as follows:
>
> ```markdown
> We are working with a tabular dataset.
> Your task is to develop a step-by-step plan to answer a query on a given table.
>
> ### Table:
> /*
> col : id | name | score
> row 1 : 1 | alice | 85
> row 2 : 2 | bob | 90
> row 3 : 3 | charlie | 75
> */
> Query: what is the score of alice?
> Plan:
> 1. Select rows where the 'name' is 'alice'.
> 2. Select the 'score' column.
> ```
>
> We ran POS again on both datasets using these generalized in-context examples. The results indicate that this change led to only a slight decrease in accuracy compared to dataset-specific examples, as shown in the table below:
>
> Table: Table QA accuracy (&uarr;) on TabFact and WikiTQ datasets using general in-context examples for planning.
>
> |                             Method                              | TabFact | WikiTQ |
> |:---------------------------------------------------------------:|:-------:|:------:|
> |                          End-to-end QA                          |  71.17% | 49.24% |
> |         POS with 5 general in-context planning examples         |  81.08% | 50.92% |
> |       POS with 30 (full) general in-context planning examples        |  81.92% | 52.00% |
> |   POS with 30 (full) dataset-specific in-context planning examples   |  83.45% | 54.95% |
>
> These numbers confirm that POS remains competitive without a strong reliance on
> dataset-specific examples (`1.5%` → `3.0%` accuracy gaps). Additionally, we observed that the general in-context
> examples led to low unprocessable rates, with TabFact rates changing from
> `3.16%` → `3.06%` and WikiTQ rates from `13.58%` → `14.48%`.
>
> Finally, it is also worth noting that reliance on tailored, dataset-specific examples is common across other methods, such as CoTable.
> Typically, general prompts (e.g., Meta-prompting[1]) yield slightly lower accuracy than dataset-specific prompts but improve generalizability, a trend that aligns with our findings.
>
> - [1] Meta-Prompting: Enhancing Language Models with Task-Agnostic Scaffolding

---

> ### Author Response · Authors · 2024-11-14
> **Regarding the extension of evaluation benchmarks**
>
> > 4. Further experimentation across a broader range of datasets (ideally 5–6) would be needed to demonstrate POS’s generalizability and robustness.
>
> Thank you for this suggestion!
>
> We would like to note that TabFact and WikiTQ are also the primary evaluation datasets
> in existing works, including our main baselines—CoTable, Binder, Dater, and TableSQLify.
>
> Due to budget and time constraints, we are currently unable to extend POS testing to
> additional datasets as suggested. The cost of running POS on TabFact and WikiTQ is
> particularly high due to the large input tables and the in-context examples required
> for each prompt, resulting in a high token usage. For instance, a single evaluation on
>
> TabFact or WikiTQ using step-by-step planning (in Question 1) incurred over `$600` in API costs.
> Given these constraints, we prioritized
> TabFact and WikiTQ as they offer a robust and feasible evaluation within our available
> resources.
>
> If there are any additional questions or further concerns, we would be more than happy to address them and continue the discussion. We are eager to engage in any way!
>
> Thank you once again for your time and consideration.

---

> ### Comment · Reviewer_j2Vr · 2024-11-17
> **Thank you for the detailed reply from the authors**
>
> Thank you for the detailed reply from the authors. Although I still have concerns with regards to the novelty and the evaluation datasets, the new results have resolved some of my concerns regarding the unprocessable rate and the generalizability of in-context examples. So I decide to raise my score from 3 to 5.

---

> ### Author Response · Authors · 2024-11-23
> **Extending POS to Free-form Table QA**
>
> Dear Reviewer `j2Vr`,
>
> As promised, we extended POS (our method) to **FeTaQA (Free-form Table Question Answering)** dataset. On this dataset, we compared POS with End-to-End QA and Few-Shot QA, as we did with TabFact and WikiTQ. We report the results as follows:
>
> Table: Evaluation on FeTaQA test set with gpt4o-mini.
> | Method             |    BLEU   |  ROUGE-1 |  ROUGE-L  |
> |--------------------|:---------:|:--------:|:---------:|
> | 0-shot QA   |   18.99   |   51.92  |   46.44   |
> | Few-shot QA |   19.18   |   53.32  |   46.86   |
> | Plan-of-SQLs (POS)      | **20.16** | **54.7** | **48.69** |
>
> We find that POS is better than End-to-End or Few-Shot QA in both aspects, accuracy and interpretability:
> - POS consistently achieves the best scores across all metrics on the FeTaQA test set, with a BLEU improvement of 0.98 pts and ROUGE-L gain of 1.83 pts over Few-shot QA. The reason for these improvements is that POS retrieves fine-grained and correct information from tables through its step-by-step SQLs, enabling better answer generation. Here we show an example of precise table information retrieval from POS.
>
> ```python
> Table Page Title: 1948 Oregon gubernatorial special election
> Table Section Title: Results
> ### Table:
> /*
> col : Party | Party | Candidate | Votes | %
> row 1 : - | Republican | Douglas McKay | 271,295 | 53.23
> row 2 : - | Democratic | Lew Wallace | 226,958 | 44.53
> row 3 : - | Independent | Wendell E. Barnett | 11,380 | 2.23
> row 4 : Total votes | Total votes | Total votes | 509,633 | 100
> row 5 : - | Republican hold | Republican hold | Republican hold | Republican hold
> */
> Question: By what percentage of votes did McKay win the election against Wallace's 44.53% vote?
> Plan:
> 1. Select rows where 'Candidate' is 'McKay' or 'Wallace'.
> 2. Select 'Candidate' and '%' columns.
> 3. Ask LLM to generate the final answer based on the current table.
> ```
> This information retrieval contrasts with End-to-End QA and Few-shot QA, which feed the entire large table into the LLM, making it harder for model to pinpoint relevant information and increasing the likelihood of errors.
>
> - POS provides visual explanations for the generated answers, which End-to-End QA and Few-shot QA lack. **We share an explanation from POS on FetaQA** [here](https://docs.google.com/presentation/d/1sefWsjA664LP9uhKAqDV2EtDpivK7O8h/edit?usp=sharing&ouid=106754253201604804412&rtpof=true&sd=true). As POS bases on a concise, final table to generate the final answer, this makes it easier for users to understand why the model generates a particular response. In contrast, End-to-End QA and Few-shot QA require users to refer back to the large, original table, making the reasoning process less understandable and harder to follow.

---

### Note · Authors · 2024-12-04

**Comment:**

After reviewing the feedback, we have decided to withdraw the paper to refine our work.

Thank you all for your time and consideration!

**Withdrawal Confirmation:**

I have read and agree with the venue's withdrawal policy on behalf of myself and my co-authors.